# Molecular choreography of primer synthesis by the eukaryotic Pol α-primase

Zuanning Yuan [1], Roxana Georgescu[2], Huilin Li [1] ✉ & Michael E. O'Donnell [2] ✉

The eukaryotic polymerase α (Pol α) synthesizes an RNA-DNA hybrid primer of 20–30 nucleotides. Pol α is composed of Pol1, Pol12, Primase 1 (Pri1), and Pri2. Pol1 and Pri1 contain the DNA polymerase and RNA primase activities, respectively. It has been unclear how Pol α hands over an RNA primer from Pri1 to Pol1 for DNA primer extension, and how the primer length is defined. Here we report the cryo-EM analysis of yeast Pol α in the apo, primer initiation, primer elongation, RNA primer hand-off from Pri1 to Pol1, and DNA extension states, revealing a series of very large movements. We reveal a critical point at which Pol1-core moves to take over the 3'-end of the RNA from Pri1. DNA extension is limited by a spiral motion of Pol1-core. Since both Pri1 and Pol1-core are flexibly attached to a stable platform, primer growth produces stress that limits the primer length.

DNA polymerases can only add deoxynucleotides (dNTP) to the 3' hydroxyl group of an existing primer annealed to a DNA template, and therefore priming is a prerequisite of DNA replication in all kingdoms of life[1]. Primers of bacteria and bacteriophages are composed entirely of RNA 4-12 nucleotide (nt) long that are synthesized by single-polypeptide primases[2,3]. In contrast, eukaryotic primers are RNA-DNA hybrids that are 20–30 nt long produced by the Pol α heterotetramer[4–8]. Isolation and characterization of Pol α date back nearly four decades ago[6,7,9,10]. The four-subunit Pol α was surprisingly found to be a dual-function complex with both RNA primase and DNA polymerase activities[6–8], and is often referred to as Pol α-primase. It is now well established that primer synthesis involves at least three key steps: (1) RNA primer synthesis by the primase, (2) intramolecular handover of the RNA primer to the DNA polymerase, and (3) limited DNA extension by the DNA polymerase[5,11,12].

The *S. cerevisiae* catalytic primase 1 (Pri1, p49, PriS, or PRIM1 in human) and the regulatory subunit, primase 2 (Pri2, p58, PriL or PRIM2 in human) form a heterodimer to synthesize -10 nt RNA primers[13]. and the catalytic DNA polymerase 1 (Pol1, p180 or POLA1 in human) and regulatory Pol12 subunit (p70 or POLA2 in human, also called the B-subunit) work together to extend the RNA primer by a limited DNA segment to yield RNA-DNA primers of around 25–30 nt (Fig. 1a)[14]. The self-limiting action of Pol α-primase is important because it lacks a proofreading 3'–5' exonuclease and therefore the DNA portion lacks high fidelity and appears to be proofread by Pol δ[15]. The two hetero-dimers (Pri1–Pri2 and Pol1–Pol12) assemble a constitutive Pol α-primase complex that is sometimes also referred to as a primosome[16–18]. The chimeric RNA-DNA primers produced by Pol α-primase initiate both stands, and thus are presumed to be used by both leading strand DNA polymerase Pol ε and lagging strand DNA polymerase Pol δ to duplicate genomic DNA[19]. However, more recent data suggest that Pol δ sometimes acts on the leading strand for initial extension of the hybrid primer[20,21].

The *S. cerevisiae* DNA polymerase catalytic subunit Pol1 (amino acid (aa) 1–1468) can be divided into the intrinsically disordered N-terminal domain (Pol1-NTD; aa 1–348) that is dispensable for polymerase activity but interacts with other proteins, such as Cdc13[22] and histone H3 and H4[23], the central catalytic core (Pol1-core; aa 349–1259), and the C-terminal domain (Pol1-CTD, aa 1260–1468). The Pol1-core structure adopts the universal right-handed DNA polymerase fold consisting of an extended N-terminal region (N-term), a catalytic palm domain, a helical fingers domain that interacts with the incoming dNTP, and a thumb domain that binds and stabilizes the template/primer (T/P) substrate[24–27]. The Pol1-CTD contains two Zn-binding motifs linked by a three-helix bundle[14]. The polymerase accessory subunit Pol12 (aa 1–705) contains an N-terminal

[1]Department of Structural Biology, Van Andel Institute, Grand Rapids, MI, USA. [2]DNA Replication Laboratory and Howard Hughes Medical Institute, Rockefeller University, New York, NY, USA. ✉e-mail: Huilin.Li@vai.org; odonnel@rockefeller.edu

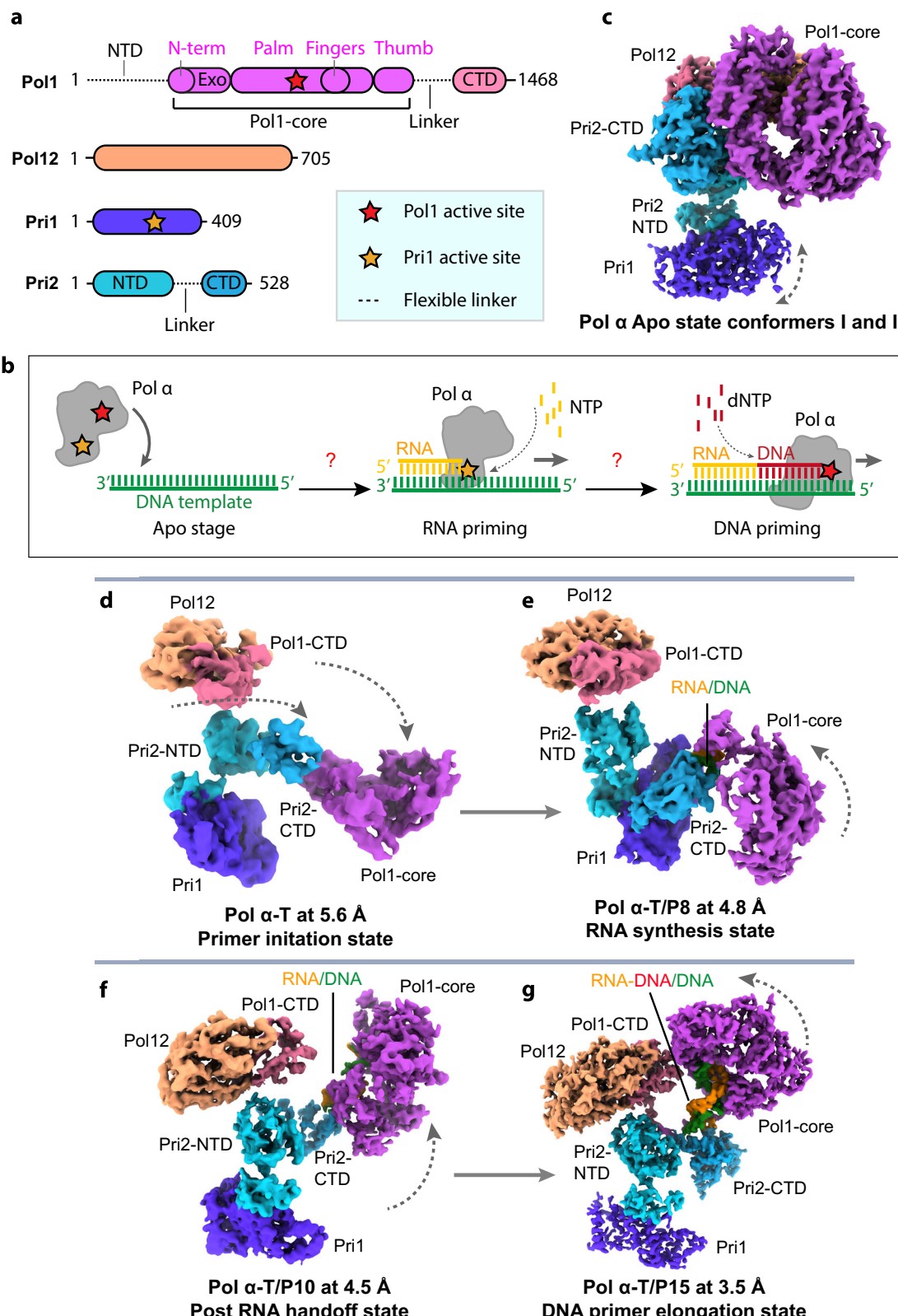

**Fig. 1 | Cryo-EM structures of the S. cerevisiae Pol α. a** Domain architecture of the four subunits of the holoenzyme. The red asterisk indicates the polymerase activity in the catalytic NTD of Pol1 which is referred to as Pol1-core. The yellow asterisk indicates the primase activity in Pri1. **b** A sketch of Pol α RNA primer and DNA primer synthesis steps. **c**–**g** Six cryo-EM maps of Pol α in different states of catalysis: Pol α in the absence of T/P exists in the Apo state (conformers I and II), **c**), Pol α–T (T refers to template DNA) is in the primer initiation state (**d**), Pol α–T/P8 (T/P8 refers to the 8-nt RNA primer annealed to the template) is in the RNA synthesis state (**e**), Pol α–T/P10 (T/P10 refers to the 10-nt RNA primer annealed to the template) is in the post RNA hand-off state (**f**), and Pol α–T/P15 (T/P15 refers to the 10-nt RNA and 5-nt DNA chimeric 15-mer primer annealed to the template) is in the DNA elongation state.

oligonucleotide/oligosaccharide (OB) domain (Pol12_NTD) and a C-terminal catalytically inactive phosphodiesterase (PDE) domain (Pol12-CTD). Pol12 interacts with Pol1-CTD to assemble the Pol1–Pol12 heterodimer[14]. Interestingly, a 10-bp RNA/DNA T/P duplex bound to the Pol1-core was found to be in the A form, suggesting that the ensuing B-form DNA/DNA T/P duplex would bind less well to the Pol1-core, leading to the proposal that a weakened binding of the DNA/DNA T/P region enables the spontaneous release of the completed RNA-DNA primer and the termination of DNA extension[24]. An alternative model suggests that flexible linkers that bind Pri2 and Pol1/12 constrain primer length[28]. The current report adds to these views and proposes a possible structure-based explanation that might limit extension of the RNA-DNA hybrid primer.

The catalytic Pri1 has a "prim" fold with an inserted all-helical domain[29]. The prim fold is conserved in the prim-pol superfamily and has a Pac-Man-like shape, with two small β-sheets forming the upper and lower jaws that are surrounded by α-helices on the outside. Pri2 is divided into a globular N-terminal helical domain (Pri2-NTD) and an elongated C-terminal 12-α-helices domain (Pri2-CTD) that coordinates a 4Fe-4S cluster[29–32]. Inserted in the Pri2-NTD is a small α/β subdomain that mediates interactions between Pri1 and Pri2. Interestingly, the Pri2-NTD was shown to interact with the C-terminal 18-residue hydrophobic peptide of Pol1, establishing a physical link between the primase and the polymerase within the Pol α complex[29,32].

The above structural knowledge was derived from studying Pol α domains and sub-complexes. Because the primase module and the polymerase module are flexibly tethered[33], it is inherently challenging to study the structure of the full complex. The first milestone toward this goal was the determination of the crystal structure of the human Pol α in an apo state, revealing a compact architecture in which the DNA polymerase catalytic core p180-core (Pol1-core) is in an inhibited state and a central role of p58-CTD (Pri2-CTD) in coordinating the two catalytic domains[16]. Pol α can be recruited by the shelterin complex and be stabilized by the telomere single-strand overhang DNA binding protein complex CTC1-STN1-TEN1 (CST) to perform C-strand fill-in synthesis. A recent 4.6 Å cryo-EM structure of the human Pol α bound to the CST in the presence of a telomeric ssDNA showed that Pol α is in a state that is similar to the apo crystal structure[34]. However, a similar study of the CST–Pol α bound to telomere templates captured the human Pol α in a primer initiation state in which polymerase and primase are separated by the CST but both bind to the template DNA[35]. Another study of *Tetrahymena* Pol α in complex with the CST and telomerase complexes also captured the Pol α in an active state in which the polymerase and primase are separated by the CST[36]. Due to the presence of the CST and/or telomerase in these studies, it is unclear if Pol α takes on these reported poses during normal DNA replication in the absence of CST/telomerase. Finally, the human Pol α structure bound to a DNA template annealed to the 12 nt RNA-DNA primer was recently reported, revealing the Pol α configuration in the DNA synthesizing mode[37].

These recent advances have greatly enriched our understanding of the Pol α-primase. However, a detailed description of the primer production mechanism is still missing, in particular how Pol α-primase passes the RNA primer from the primase to the DNA polymerase (Fig. 1b). Specifically, what has been lacking is a systematic study of all the major stages of primer synthesis with an intact Pol α-primase complex—without truncations or other binding partners. We describe in this report the full Pol α-primase complex acting on a series of T/P substrates that vary in primer length to determine the holoenzyme structure in the apo state, the primer initiation state, the post primer hand-off state, and the DNA extension state. Our study elucidates a series of large-scale conformational changes that Pol α-primase undergoes to synthesize the hybrid RNA-DNA primer, revealing the RNA hand-off mechanism, and suggesting determinants by which the primer length is limited.

## Results

### Cryo-EM of Pol α-primase in apo, primer initiation, RNA synthesis, primer hand-off, and DNA elongation states

We purified the Pol α-primase complex and solved its structure using cryo-EM either alone or with several particular nucleic acid substrates (Supplementary Fig. 1). We found the apo complex existed in two major conformations and determined their structures to 3.7 Å and 3.8 Å resolution and refer to them here as Apo state conformers I and II, respectively (Fig. 1c, Supplementary Figs. 1 and 2 and Supplementary Table 1). To visualize how the apo enzyme transitions to engage a template DNA, we added a 60-nt template ssDNA (T) into the purified Pol α-primase at a molar ratio of 1:1 and determined the structure of Pol α-primase–Template DNA at 5.6 Å resolution (referred to here as the primer initiation state (Fig. 1d, Supplementary Figs. 1c and 3 and Supplementary Tables 1 and 2). To capture the Pol α-primase structure in an RNA-primer synthesizing pose we examined mixtures of Pol α-primase with a DNA template paired with an RNA primer of 6, 7, or 8 nt respectively (Supplementary Table 2 and Fig. 1d, e). For convenience, we refer to the substrate as T/Px with X referring to the number of nt in the primer. Cryo-EM study showed that Pol α–T/P6, Pol α–T/P7, and Pol α–T/P8 are of essentially the same architecture. Therefore, we chose to present the Pol α–T/P8 structure that was determined to the best resolution of 4.8 Å and assigned it to the RNA synthesis state; the longer primer may have better stabilized the structure (Fig. 1e and Supplementary Figs. 1d and 4).

Although a typical eukaryotic RNA primer is 8 to 10 nt long, the structure of Pol α–T/P8 showed that the 8-nt RNA primer bound to the template (T/P8) did not trigger primer hand-off from the primase (Pri1) to the polymerase (Pol1-core). We next examined several T/P substrates with increasingly longer RNA primers, including T/P9 (9-nt RNA primer), T/P10 (10-nt RNA primer), and T/P11 (11-nt RNA primer) (Supplementary Table 2). Cryo-EM showed that an RNA primer with a minimal length of 10 nt paired with the template DNA (T/P10) triggered the primer hand-off to yield a post RNA hand-off state. We determined a cryo-EM map of the Pol α–T/P10 complex at 4.5 Å resolution in the post RNA hand-off state (Figs. 1f and 4 and Supplementary Figs. 1e and 5). Finally, to visualize Pol α-primase in the DNA primer extension mode, we coupled the template DNA with a 15-nt RNA/DNA chimeric RNA-DNA primer, composed of a 10-nt RNA segment and a 5-nt DNA segment. Cryo-EM analysis of the Pol α–T/P15 complex stabilized by 1 mM ddCTP (i.e., next nucleotide added) led to a 3.5 Å EM map of the enzyme in the DNA synthesizing pose, which is termed the DNA elongation state (Fig. 1g and Supplementary Figs. 1f and 6). Figure 1d–g panels are shown relative to a fixed Pol12-Pol1-CTD platform to define a "standard view, and the correlation coefficients between the model and maps are given in Supplementary Table 3.

These structures reveal a modular architecture of Pol α-primase, in which both Pol1 and Pri2 are separated into their respective flexibly linked NTD and CTD. The linker between Pol1-core and Pol1-CTD is 10-residues long, and the linker between Pri2-NTD and Pri2-CTD is 20-residues long. The noncatalytic Pol1-CTD and Pol12 assemble to form a stable "platform", and the Pri2-NTD and Pri2-CTD together function as a flexible hinge. The platform (Pol1-CTD–Pol12) and the hinge (Pri2-NTD–Pri2-CTD) then work together to enable and coordinate large conformational changes of the RNA primase Pri1 and the DNA polymerase Pol1-core during primer synthesis.

### The polymerase site is blocked in apo Pol α-primase

The apo Pol α-primase is elliptic and measures $150 \times 120 \times 110$ Å. In the apo structure, Pol1 and Pol12 are fully engaged with a buried interface of 202 Å², in which Pol12 and Pol1-CTD form the "platform", and Pol1-core tightly binds this platform, such that the OB domain of Pol12 blocks the Pol1-core from binding a polynucleotide substrate from the top, and the Pri2-CTD binds the Pol1-core thumb and palm subdomains from the left side, apparently preventing access of a DNA template to

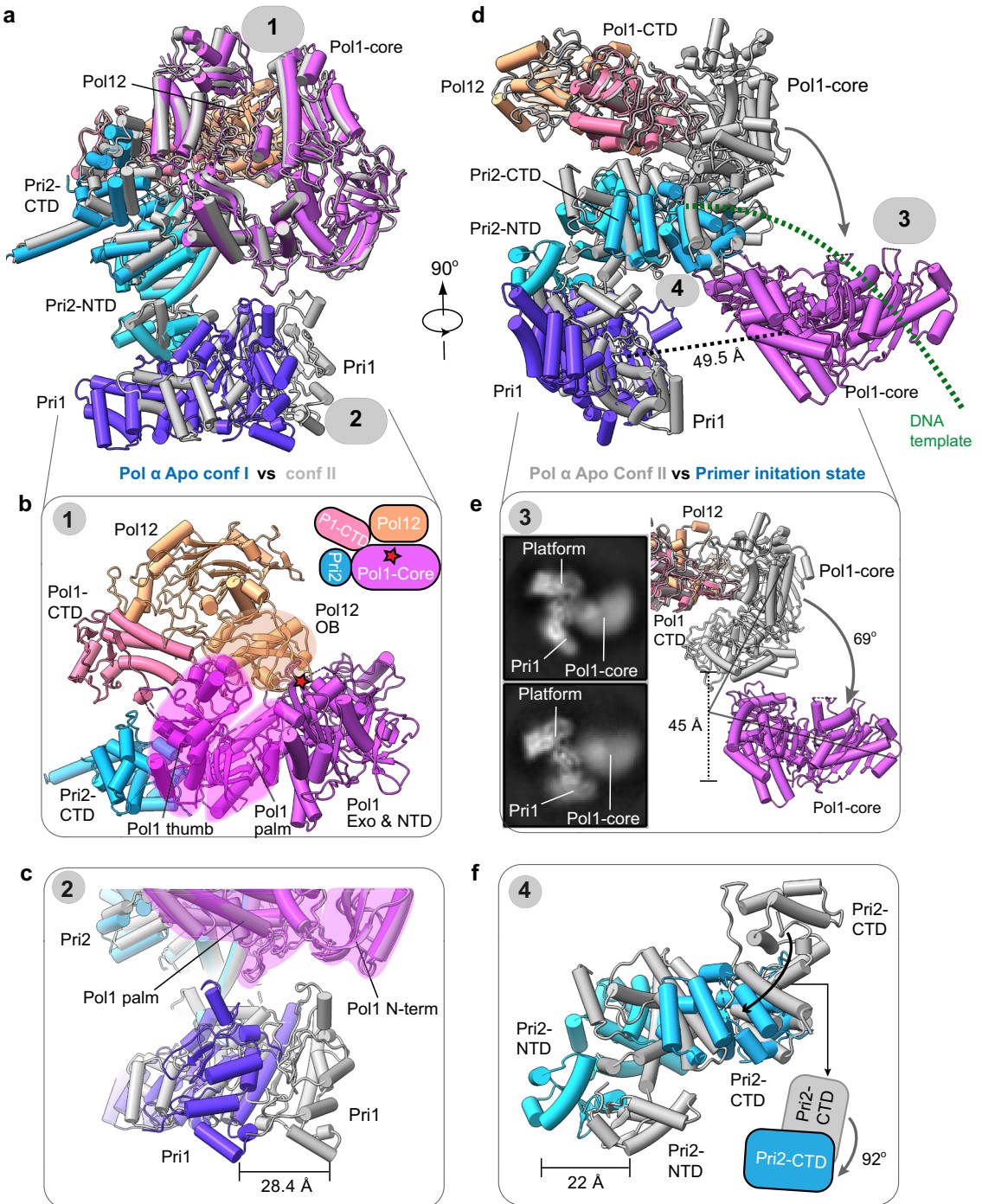

**Fig. 2 | Atomic models of Pol α in the apo and primer initiation states. a–c** Two conformations of the apo state of Pol α-primase. **a** Front cartoon view of super-imposed Pol α-primase apo state conformers I and II. Conformer I is in color and II is in gray. **b** Enlarged view of the interaction between Pol1-core, Pri2-CTD, and Pol12/Pol1-CTD platform, the catalytic site of Pol1 is indicated by the red asterisk. **c** Comparison of Pri1 between apo conf I and II. Pri1-NTD binds to different domains of Pol1: Pri1-NTD binds to the Pol1 palm in apo conf I but to the N-term of Pol1-core the Pol1-core catalytic site (Fig. 2a, b). The primase Pri1 is rectangular and connects to the platform via the Pri2-NTD hinge. Due to the small interface with the hinge, Pri1 is partially mobile and fluctuates between two states that are 28.4 Å apart: in apo conformer I, the Pri1-NTD contacts the N-term of Pol1-core, and in apo conformer II, Pri1-NTD connects to the Pol1-core palm instead (Fig. 2c and Supplementary Movie 1). The template DNA binding site in Pri1 is exposed, suggesting

in apo conf II. **d** Side cartoon view of superimposed Pol α-primase in Apo conf II and primer initiation state. Pol1-core is dissociated from the Pol12/Pol1-CTD platform. **e** Comparison of Pol1-core between apo conf II and primer initiation state: Pol1-core rotates 69° downwards toward the primase. The inset shows representative 2D averages in the initiation state, showing the partially flexible Pol1-core and the Pri1 movement. **f** Comparison of Pri2-CTD between apo conf II and primer initiation state. Pri2-CTD shifts 22 Å and rotates 92°.

the apo enzyme is ready to engage the DNA template by the primase but with its polymerase site being autoinhibited. The yeast Pol α-primase apo structures I and II resemble the published crystal structure of the human apo Pol α-primase (5EXR): the structures of the Pol1-core and Pri2-NTD are largely superimposable while the Pri1 region is somewhat displaced in the solution cryo-EM structure of the yeast Pol α-primase from the human Pol α-primase (Supplementary

Fig. 7a–c). Despite these differences, the polymerase site is largely blocked and as such, a blocked site is apparently a conserved feature of the eukaryotic apo Pol α-primase complex.

### Pri1 and Pol1-core engage the template DNA at a distance in the primer initiation state

Upon binding template ssDNA, the Pol1-core dissociates from the platform, leading to a drastically different Pol α-primase structure to form the template-bound primer initiation state (Fig. 2d–f, Supplementary Figs. 3 and 8a–c and Supplementary Movie 1). The two-lobed elliptic apo structure now becomes three-lobed with each lobe being loosely associated with the central hinge, as evidenced by 2D class averaged images in which the position and orientation of the Pol1-core lobe, the platform lobe, and the Pri1 lobe vary (Fig. 2e and Supplementary Fig. 8a). The significant conformational variability limited the achieved resolution of the EM map to 5.6 Å, insufficient to resolve the bound template ssDNA (Fig. 1d). However, the overall architecture of the primer initiation state of yeast Pol α-primase is similar to a recently reported structure of the human Pol α−T (ssDNA) preinitiation complex stabilized by the telomeric CST complex such that the template DNA was resolved, although the detailed structures differ[36] (Supplementary Fig. 8d). Therefore, the ssDNA bound to human Pol α-primase is used as a placeholder for the template DNA in the yeast Pol α-primase (Fig. 2d and Supplementary Fig. 8c, d).

Transitioning from the apo state, the ssDNA template bound form of Pol1-core (i.e., state III) drops down 45 Å from the Pol12−Pol1-CTD platform and rotates 69° for binding the template DNA (Fig. 2e and Supplementary Movie 1). Accompanying the Pol1-core movement, the Pri2-CTD rotates 92° clockwise toward the Pri1 active site for binding the template DNA, and the Pri2-NTD shifts 22 Å outwards accordingly (Fig. 2f). Therefore, both Pri1 and Pol1-core engage the template DNA, but their catalytic sites are ~49.5 Å apart (Supplementary Fig. 9). In human Pol α, the transition from the apo to the nucleotide synthesis state(s) also involves Pri2-CTD moving toward the Pri1 active site and Pol1-core dissociating from the platform toward Pri1[12,18]. Thus, the human and yeast enzymes share a drastic conformation change in the transition mechanism to initiate primer synthesis.

### The RNA synthesis state of the yeast Pol α-primase

The 4.8 Å structure of the Pol α−T/P8 complex represents the enzyme pose nearing the end of RNA priming (Figs. 1e and 3a and Supplementary Fig. 4). During RNA synthesis, the overall architecture is configured similarly to the initiation state bound to template DNA. The Pri1 and platform (Pol1-CTD−Pol12) form an axis on one side, while Pol1-core, Pri2-CTD, and template DNA swing around this axis on the other side (Supplementary Fig. 8c–e). Although all subunits undergo conformational changes during the synthesis of the first 8 nt of the RNA primer, for simplicity, we will focus only on the three subunits directly involved in RNA synthesis and use Pri1 as the reference to align these three subunits (Fig. 3c–f). Both Pri2-CTD and Pol1-core have undergone significant conformational changes to accommodate the ~2/3 helical turn of the nascent RNA/DNA duplex (Fig. 3a, b). In fact, Pri2-CTD and Pol1-core now form an open chamber with Pri1 to surround the T/P8 duplex region. The Pri2-CTD holds the primer's distal 5′ end in place, while the proximal 3′ end is positioned correctly to point to the Pri1 active site for further ribonucleotide addition. Importantly, the Pol1-core is now nearly parallel to Pri1 and assumes a position right next to the T/P8, with the thumb subdomain and the polymerase active site both approaching the primer substrate (Fig. 3b).

Compared to the primer initiation state as well as with the preinitiation state of the human Pol α−ssDNA in which Pri2-CTD or the homologous p58-CTD binds the template DNA and bridges the gap between the primase site and polymerase site (Fig. 3c), transition to the RNA synthesis state (after synthesizing an 8-nt RNA primer) involves the Pri2-CTD turning and moving downwards by 30 Å to

accommodate the 8-bp RNA/DNA duplex (Supplementary Fig. 10a). The Pri2-CTD binding site to the 5′-end of the duplex is ~40 Å from the Pri1 catalytic site (Supplementary Fig. 10b). This distance is equivalent to the length of one full-turn of A-form RNA/DNA duplex that is the average RNA primer length[24,38]. Importantly, the catalytic Pol1-core in the RNA synthesis state is poised to capture the RNA/DNA duplex and take over from the primase Pri1 (Fig. 3b–f).

We can assume the 60-bp ssDNA of the initiation state positions into the Pri1 catalytic site for RNA synthesis. Thus, the Pri1 forms an RNA primer, represented as the 8-nt RNA priming state, and the Pol1-core becomes aligned to take over the RNA primed site to form the hybrid RNA-DNA primer (Fig. 3a–f). During RNA primer handover to the Pol1-core, the Pol1-core appears to become nearly parallel with the Pri1 primase on the opposing side of the T/P8 (Fig. 3a). This face-to-face arrangement of the Pri1 and Pol1-core likely facilitates the Pol1-core to take over the RNA primer 3′ end from the Pri1 in a subsequent step. We note that the thumb of Pol1-core is flexible and not resolved in the initiation state of both the human Pol α−ssDNA and the yeast Pol α−T, but the thumb is clearly observed in the RNA synthesis state (Supplementary Fig. 10a) and thus is perhaps stabilized by the T/P8 duplex region in the post RNA hand-off state indicating an incremental formation of the active DNA polymerase pocket in this intermediate.

### RNA hand-off of the T/P10 from the RNA primase to the DNA polymerase

Although the 8-nt long RNA primer meets the minimum length requirement of the DNA polymerase[39], the less-than-a-full-turn helical duplex in the 8 nt RNA synthesis state structure was apparently insufficient to trigger the primer hand-off from the primase to the polymerase. As we mentioned above, it took a 10-nt RNA primer to trigger the hand-off in our in vitro system, as revealed by the 4.5 Å resolution structure of the Pol α−T/P10, a post RNA hand-off to DNA Pol1 state (Figs. 4a and 1f and Supplementary Fig. 5). The post RNA hand-off state structure shows that the primer 3′ end of the T/P10 is engaged with the DNA Pol1 active site and that the Pri1 primase is fully dissociated from the RNA-DNA P/T. The structure is still three-lobed, with the Pri2-NTD and Pri2-CTD hinging together the Pol1-CTD−Pol12 platform (lobe 1), the Pri1 (lobe 2), and the Pol1-core−T/P10 (lobe 3) (Fig. 4a).

Alignment of the 8 nt RNA synthesis state and RNA hand-off to DNA Pol1 state structures by superimposing the T/P10 and associated Pri2-CTD reveals a large-scale 60° rotation of Pol1-core from touching the side of the T/P8 to fully engaging the T/P10 3′ end-on in the catalytic site (Fig. 4b–d). In the post RNA hand-off to Pol1 state, the thumb and palm fit in the two minor grooves of the T/P10 duplex and interact primarily with the RNA primer strand of the duplex, consistent with previously reported crystal structures of the truncated yeast and human Pol1 catalytic domain bound to a DNA/RNA or a DNA/DNA duplex[24,27]. In our full-length Pol α−primase structure, the thumb binds P2-5, and the palm binds P5-10 (Fig. 4a). This T/P binding mode may explain why a shorter RNA primer (6–9 nt) does not trigger the primer hand-off: a shorter primer can stably bind the thumb but not the palm, and a stable primer binding by the palm perhaps requires P5-10. We therefore speculate that only when the RNA primer reaches 10 nt can the Pol1-core catalytic pocket compete with the Pri1 and stably engage the T/P10 and be converted to the polymerase pose. In the Pol α apo state, we observed that Pol1-NTD interacts with Pri1-NTD in ~50% of the particle population, yet the Pri1 EM density is weak and broken. This indicates to us a "transient" nature of the interaction. Transitioning from Pol α-T/P8 to Pol α-T/P10, we observed that Pol1-NTD moves toward Pri1-NTD to bind the T/P10 duplex (Fig. 4b–d). Although the transition step was not captured, we hypothesize that Pol1-NTD interacts with Pri1-NTD during this transition, leading to the release of Pri1 from template DNA. We propose that during the large-scale transition of Pol1-core, the N-terminus of Pol1-core "pushes" on the Pri1-

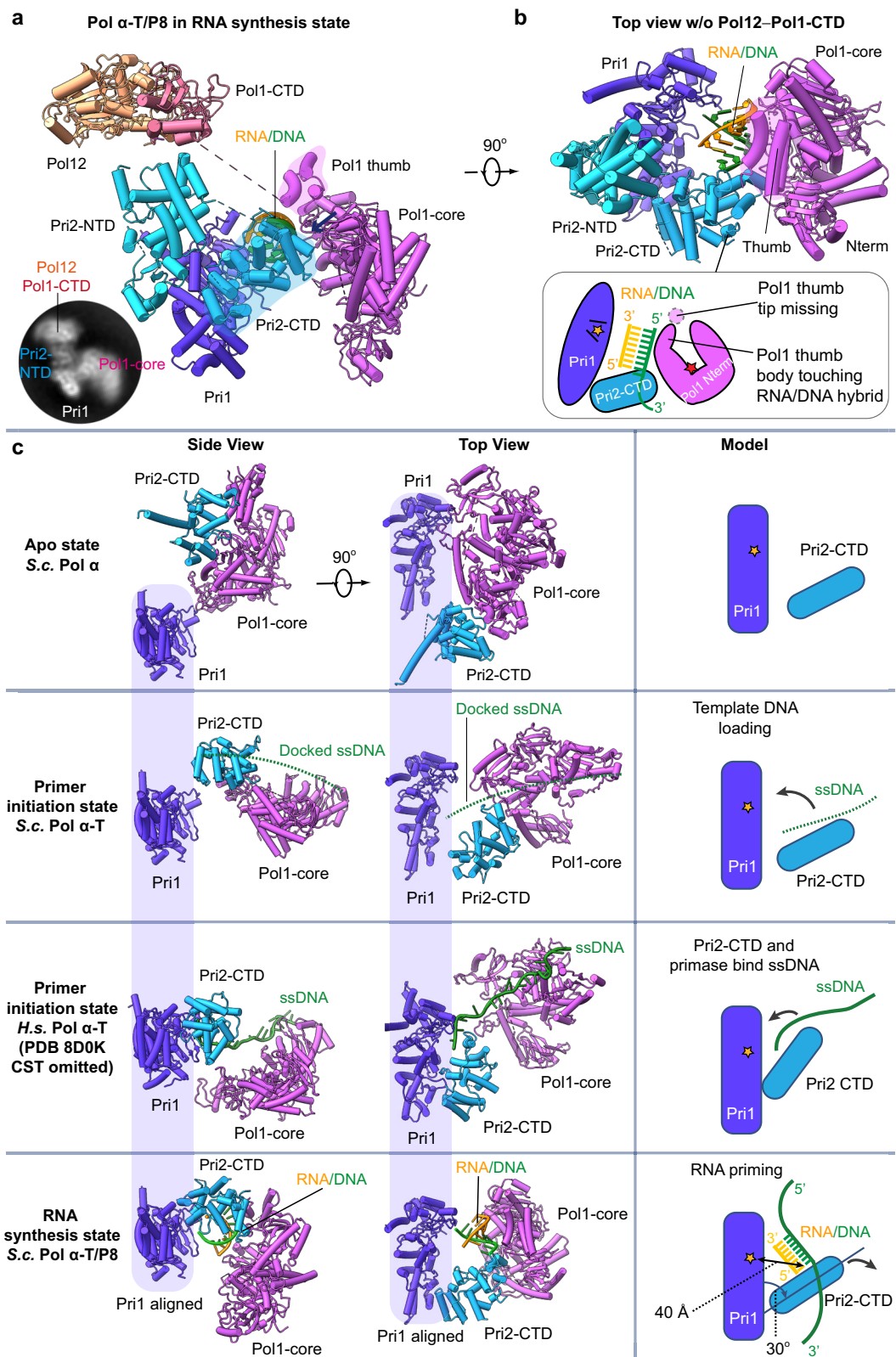

**Fig. 3 | Atomic model of Pol α−T/P8 in the RNA synthesis state. a** Front cartoon view of Pol α−T/P8. Inserted at lower left is a typical 2D class average showing the domain arrangement and the partially flexible Pol1-core. **b** Top view of Pol α−T/P8 with Pol12 and Pol1-CTD removed for clarity. **c** Comparison on how Pri1 and Pri2-CTD interact with template and primer in the apo state conf II (1st row), primer initiation state (2nd and 3rd rows), and RNA priming state (4th row). The human Pol α-primase in the 3rd row is from the PDB ([8D0K](8D0K)). These structures are shown in a side (left column) and a top (middle column) view. The right column is a cartoon model showing how Pri1 and Pri2-CTD are arranged to load template DNA and synthesize an RNA primer.

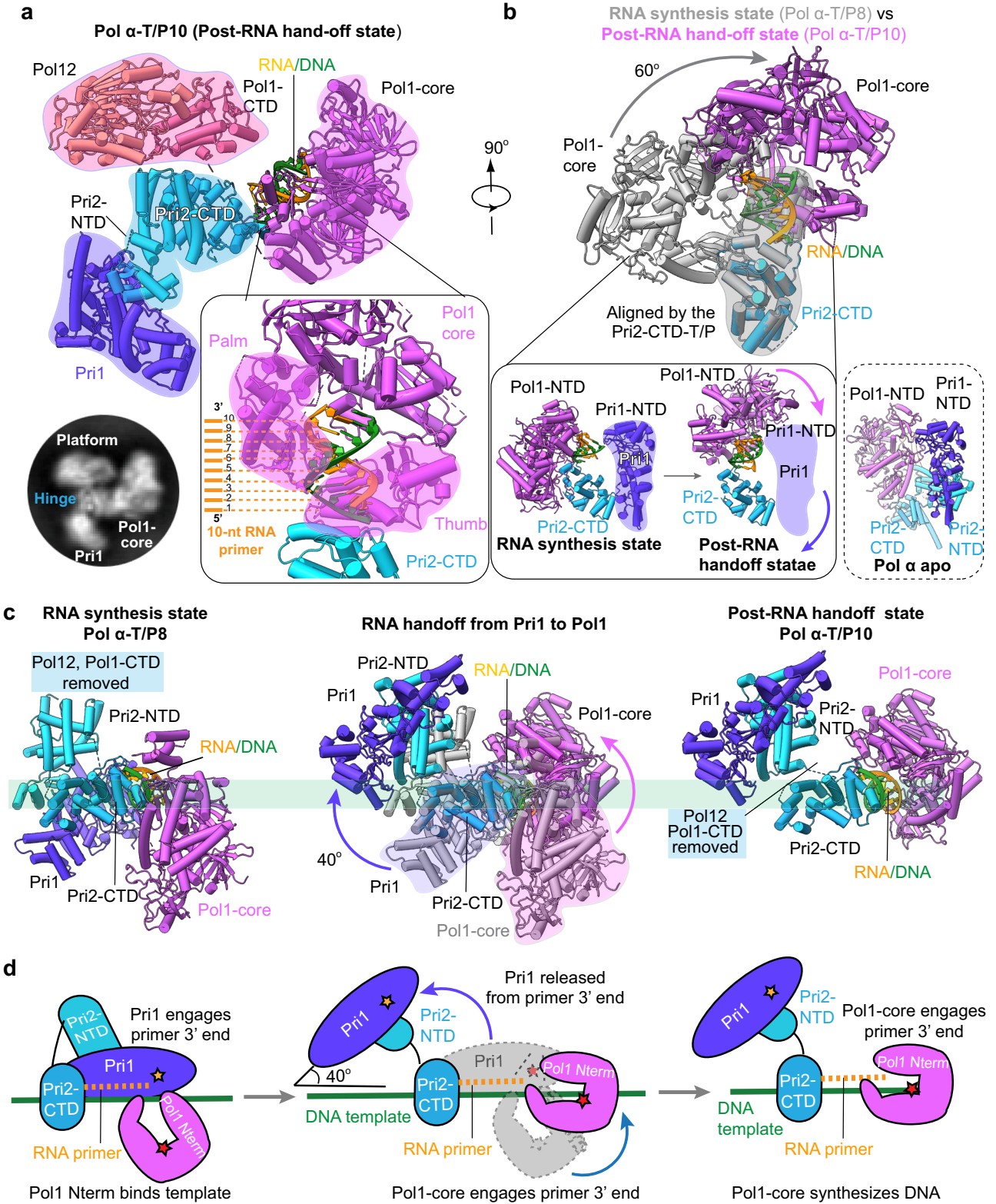

**Fig. 4 | Atomic model of Pol α−T/P10 in the post RNA hand-off state. a** Front cartoon view of Pol α−T/P10 in the post RNA hand-off state. Insert at lower left is a typical 2D class average. The lower right panel shows an enlarged view of the interface between Pol1-core and RNA/DNA. The thumb and palm subdomains bind the RNA primer nt 2–9, and the thumb is flush with P5. **b** Comparison of Pol1-core in the RNA synthesis state (Pol α−T/P8, gray) and primer hand-off state (Pol α-T/P10, color). Pol1-core rotates 60° to bind the RNA/DNA duplex. Lower left panel shows the change of Pol1-core in the post RNA hand-off state will occupy the space of Pri1. Lower right panel shows the N-term of Pol1-core may interact with Pri1-NTD as seen in Apo conf II. **c** Changes from post RNA hand-off state to DNA elongation state. Pol1-core captures the RNA/DNA while Pri1 dissociates from template DNA. **d** Cartoon model showing RNA primer hand-off from the primase (Pri1) to the polymerase (Pol1-core).

NTD to release Pri1 from the template DNA (Fig. 4b–d). This presumed "transient interaction" between Pol1-core and Pri1-NTD was also observed in the apo enzyme conformer II structure (Fig. 4b lower right panel). In the completed post RNA hand-off to Pol1 state, Pol1-core binds the T/P10 in a catalytic pose but does not interact with other protein components, perhaps liberating the DNA polymerase module for primer extension (Fig. 4b–d); the released Pri1 is connected to the Pol1-CTD–Pol12 platform via Pri2-NTD, and they form a dumbbell-like structure. The dumbbell then pivots on the Pri2-CTD and rotates counterclockwise ~40° away from the Pol1-core (Fig. 4c, d). Thus, in the "post RNA hand-off to Pol1" state, the Pol1-CTD–Pol12 platform becomes closer to the Pol1-core, perhaps enabling the Pol1-core to bind to the Pol1-CTD in the next stage. But the current pose is compatible with DNA synthesis.

### Pol α-primase in the Pol1 DNA elongation state

The 3.5-Å structure of the Pol α–T/P15 complex captures the enzyme halfway through DNA primer extension and therefore, is in a Pol 1 "DNA elongation state" (Fig. 5a, b and Supplementary Fig. 6). This complex remains a three-lobed structure with Pri2-NTD and Pri2-CTD hinging together the three separate lobes: the platform, Pri1, and the RNA/DNA T/P15-nt-bound Pol1-core (Fig. 5a). However, the relative locations of these lobes have changed compared to the post RNA hand-off state— after having extended the 10-nt RNA by 5-nt DNA by Pol1-core. If we use the Pri2-CTD bound T/P as a reference to align the two states, we find that the trajectory of the Pol1-core is a right-handed spiral. In other words, the catalytic Pol1-core tracks along and spirals around the growing tip of the T/P duplex (Fig. 5c, d). This is best shown by the position of the thumb and palm of Pol1-core: they interact with the first minor groove of the T/P8 in the post RNA hand-off state and spiral up to engage with the second minor groove of the now elongated RNA/DNA T/P15-nt in the DNA elongation state (Fig. 5c, d). Because the DNA elongation state is nearing the end of primer synthesis, the Pol1-core is now associated with the platform via a physical interaction with Pol1-CTD, perhaps preparing Pol α-primase for returning to its starting point in the apo form where Pol1-core is "parked" away on the platform.

The 5'-end and the 3'-end of the T/P are held by Pri2-CTD and the Pol1-core, respectively. It is possible that the maximum distance that these two domains can separate from each other in the confinement of the enzyme complex sets the limit on the hybrid RNA-DNA primer length. The 5'-end binding Pri2-CTD is constrained by the linker to the Pri2-NTD, and the 3'-end engaged Pol1-core is constrained by the linker to Pol1-CTD which is part of the platform. And indeed, throughout the series of conformational changes, the platform, and the Pri2-NTD remain bound and form a rigid core against which all the other components move (see Fig. 1). These flexible and thus expandable linkers might facilitate definition of the RNA-DNA primer length as explained in the "Discussion".

The interaction between the Pol1-core and T/P15 is comparable to the previously reported structures of the Pol1 catalytic domain of Pol α bound to an RNA or DNA primer[24,40]. The thumb contains three T/P binding patches: residues 1074–1077 in the thumb body, residues 1130–1150 in the thumb tip, and residues 1245–1252 in the fifth helix of the helix bundle in the thumb (Fig. 5a–d). Five positively charged residues in the thumb body (K1074, R1075, R1076, K1216, R1224) interact with nt 11–12 in the DNA primer, three residues near the thumb tip (K1132, K1135, and N1145) interact with nt 9–10 in the RNA primer, and three positively charged residues (K1247, R1250, and R1251) in the 5th helix of the helical bundle of the Pol1-core thumb domain interact with nt 1–2 of the RNA primer (Fig. 5a–d and Supplementary Fig. 11). We note that this is the first time the α-helix number 5 of the thumb is observed in a Pol1-template DNA/primer complex. Residues K1046, K1048, and K1069 on the palm domain enclose the phosphate groups of nt 13–14 next to the thumb. The Pol1-core interacts with primer strands nt 1–2 and 9–14 and the template strand nt 9–14. Pri2-CTD still binds the 5'-triphosphate group of the primer as well as the 3'-end of the template DNA, i.e., the 3' blunt end of the T/P10. Strikingly, the Pri2-CTD holds to the end of the T/P throughout the priming process spanning from the addition of the first NTPs into an RNA primer to the DNA primer elongation (Fig. 5a, b and Supplementary Fig. 11). This probably ensures that Pol α-primase never loses grip on the T/P throughout the reaction.

## Discussion

We have visualized several key intermediate states of a eukaryotic Pol α-primase in this study, revealing the large conformational changes involved in eukaryotic RNA-DNA primer production. To our knowledge, this is the first time that all three major steps of the hybrid RNA-DNA primer synthesis—RNA primer synthesis, RNA primer hand-off, and DNA extension—have been captured for the Pol α-primase complex of the same organism (Fig. 6 and Supplementary Movie 1).

### Molecular choreography during primer synthesis by Pol α-primase

Integration of the observed states enables us to elaborate a multi-step reaction cycle for primer synthesis (Fig. 6 and Supplementary Movie 1). (1) In the apoenzyme resting state (apoenzyme conformers I and II)— before Pol α-primase engages template DNA, it autoinhibits the DNA Pol1-core by sequestering it on the Pol12-Pol1-CTD "platform" but allows the RNA primase Pri1 to rotate about and search for a template (Fig. 6, diagram 1). (2) In the template-bound primer initiation state, the Pri1 captures a template DNA thereby triggering a major conformational change that leads to the release of Pol1-core from the platform and the binding of the Pol1-core to the template DNA at a position that is 50 Å upstream of Pri1 (Fig. 6, diagram 2). (3) In the RNA synthesis state, the synthesized ~8–10-nt RNA primer orients the upstream Pol1-core into a pose that is compatible with T/P binding (Fig. 6, diagram 3). (4) When the RNA primer reaches 10 nt, the enzyme enters the RNA primer hand-off state, in which Pol1-core takes over the T/P from Pri1 and engages the 3' end of the RNA primer in the catalytic pocket, leading to a large rearrangement of the structure: the Pri1 and the platform together rotate away from the T/P, but the Pol1-core rotates upwards, while they are all tethered by the Pri2-NTD-platform complex (Fig. 6, diagram 4). (5) In the "DNA elongation state", the platform and the Pri1 are relatively stable, but the Pol1-core spirals around the growing RNA/DNA helical duplex, and the Pri2-CTD remains capped on the 5'-end of the RNA primer and template DNA yet tethered to the Pol1 core and Pri2-NTD via two flexible linkers (Fig. 6, diagram 5). In a hypothetical end state of DNA elongation, we suggest that Pol1-core either collides with the Pol1-CTD, or that the linker between Pol1-core and Pol1-CTD has stretched to the limit because of the Pol1-core rotation and upwards movement around the duplex, inducing the Pol1-core to let go of the completed RNA-DNA hybrid primer (Fig. 6, diagram 6). However, alternative nucleotide counting mechanisms for primer length have been proposed earlier and will be discussed in the last section of the Discussion, below. Upon Pol α-primase dissociation from the P/T, all lobes return to their resting positions in preparation for the next round of primer synthesis.

The multiple states revealed an unexpected cycling of the distance between the primase and polymerase. The active site distance is about 70 Å in the apo state (Supplementary Figs. 2b and 9). Upon binding to a template ssDNA, the distance changes to 60 Å, and we imagine this distance will continue to shorten during RNA primer synthesis, until the RNA primer reaches 10 nt, when the two catalytic domains collide and set off the RNA T/P hand-off in which DNA polymerase latches onto the 3' end of the RNA primer and pushes the primase away from the substrate, increasing the distance between primase and polymerase to 120 Å. Following the completion of DNA priming, the Pol1-core will rebind the platform, restoring the distance between the primase and

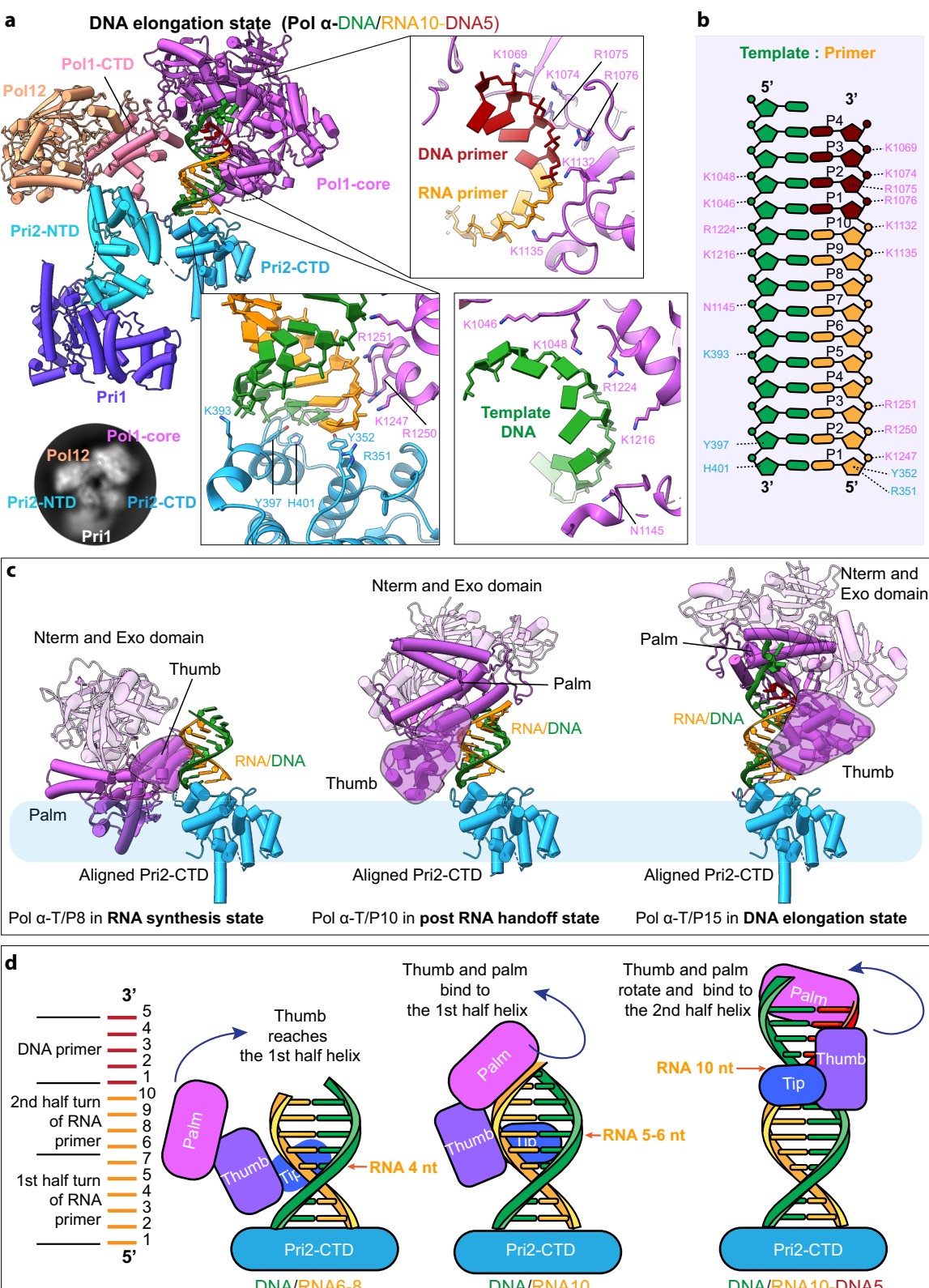

**Fig. 5 | Atomic model of Pol α−T/P15 in the DNA elongation state. a** Front cartoon view of Pol α−T/P15. Three insets show detailed contacts between Pol1-core and T/P. Inserted at lower left is a typical 2D class average showing the largely flexible Pri1. **b** Schematic diagram of the interactions of Pol α's thumb and palm subdomains with the T/P. **c** Comparison of RNA priming state, post-RNA hand-off state, and DNA elongation state. The structures are aligned based on Pri2-CTD. The Pol1-core thumb touches the bottom of the RNA/DNA duplex in the RNA synthesis state. The Pol1-core thumb and palm subdomains fit the first minor groove and interact with the RNA primer (P1–P10 region) in the post-RNA hand-off state. In the DNA elongation state, the thumb and palm subdomains of Pol1-core fit the second minor groove and interact with primer strands nt 1–2 and 9–14, as well as the template strand nt 9–14. **d** Sketch of DNA polymerase loading and DNA primer elongation. The schematic DNA drawings were modified from a DNA template (https://www.vecteezy.com/vector-art/3069633-dna-molecule-icon-vector-illustration-on-white-background).

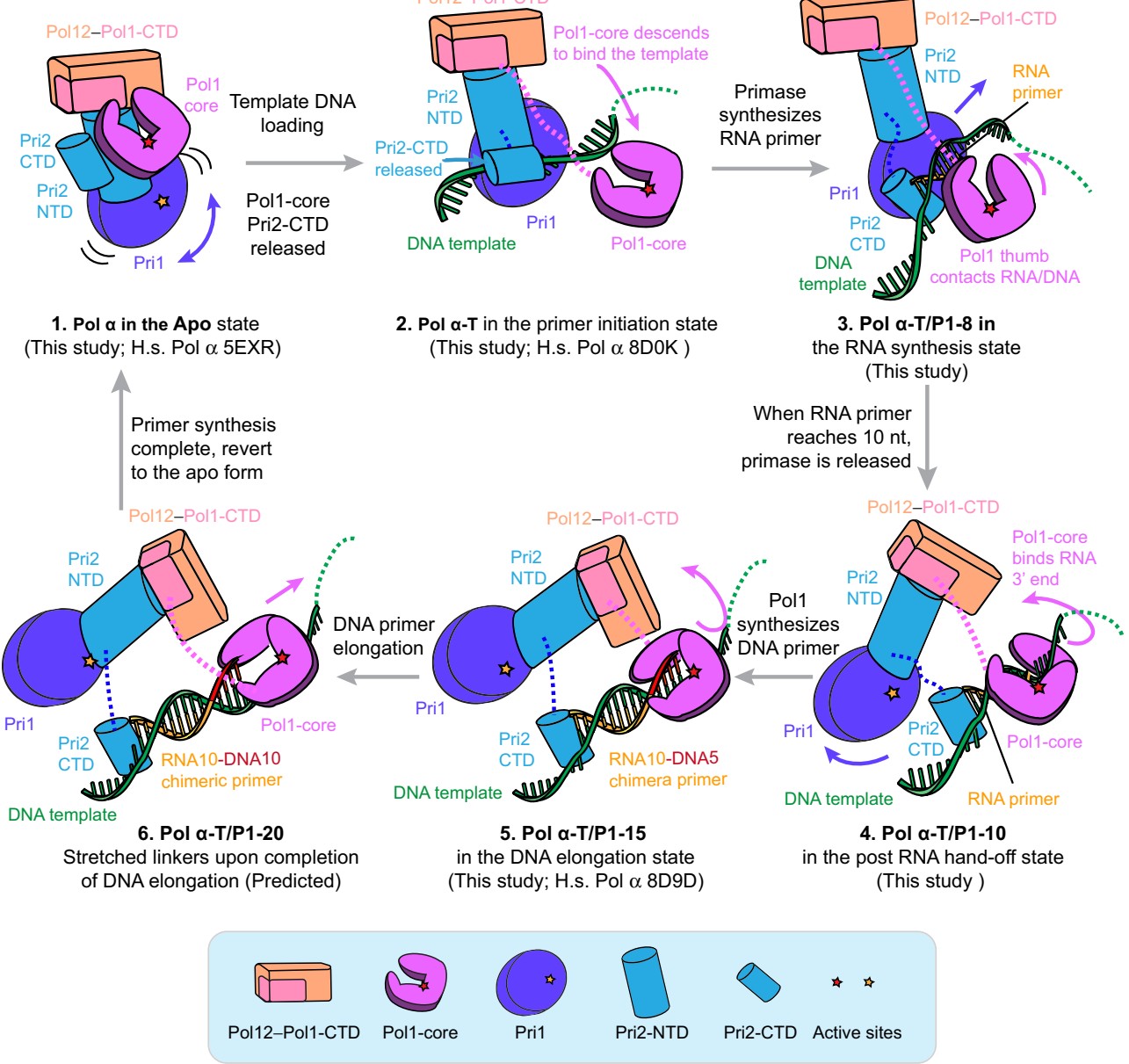

**Fig. 6 | Model for hybrid RNA-DNA primer synthesis by the yeast Pol α.** 1 Pol α fluctuates between apo conformers I and II in the absence of a DNA template, with the Pol1-core catalytic site being blocked by the Pol1-CTD/Pol12 platform and Pri2-CTD and Pri1 moving about. 2 In the primer initiation state, Pol α engages a template DNA. Pol1-core dissociates from the platform. Pri2-CTD directs the template DNA to the primase active site in Pri1. 3 In the RNA synthesis state, Pri1 synthesizes the RNA primer on the DNA template. The RNA/DNA hybrid duplex displaces Pri2-CTD from Pri1, allowing Pol1-core to engage the growing T/P. 4 In the RNA primer hand-off state, Pol1-core captures the 3'-end of the RNA primer and expels Pri1 from template DNA. This completes the RNA primer hand-off from Pri1 to Pol1-core. 5, 6 In the DNA elongation state, Pol1-core spirals around the RNA/DNA duplex to synthesize the DNA primer. The schematic DNA drawings were modified from a DNA template (https://www.vecteezy.com/vector-art/3069633-dna-molecule-icon-vector-illustration-on-white-background).

polymerase to 70 Å. The 70 → 60 → 120 → 70 Å cycling coincides with the distinct states of primer production and is likely a fundamental feature of the eukaryotic primer synthesis cycle (Supplementary Fig. 9a, b).

**The RNA T/P hand-off mechanism from Pri1 to Pol1**
Interestingly, the RNA T/P handoff mechanism to Pol 1 resembles a relay race in which the second-leg runner is positioned ahead of the first-leg runner. In the RNA synthesis state, Pol α-primase places its DNA polymerase Pol1-core on the DNA template well ahead of the primase Pri1. Pol1-core appears to initially interact with the T/P weakly via the Pol1 thumb domain. But as Pri1 lengthens the RNA primer and moves toward the Pol1-core, Pol1-core gradually rotates and inserts its thumb and palm sub-domains into the first minor

groove of the nascent RNA/DNA duplex. This action might enable Pol1-core to grip more tightly onto the T/P duplex, and this may facilitate the hand-off of the RNA T/P from the primase to the Pol 1-core polymerase and the subsequent release of the Pri1 primase (Figs. 4 and 6 fourth illustration). We note that Pol1-core apparently makes the most stable interaction with the T/P when a full helical turn (10 bp) of the RNA/DNA duplex has formed. This observation may explain the ~10-nt average length of the RNA primer[13,41,42]. Therefore, our structural study indicates that Pol1-core actively participates in the RNA primer length determination. Our hand-off mechanism is consistent with previous studies demonstrating that intramolecular RNA-to-DNA transfer occurs immediately after RNA primer synthesis and provides an explanation for how completion of

RNA primer synthesis signals the transition from primase to polymerase[39,43].

## The role of Pri2-CTD

Although it is not catalytic, the Pri2-CTD plays a critical role in primer synthesis. An important finding of this work is the revelation that Pri2-CTD is bound to the 5′ ss/ds end of the T/P at each step of the priming reaction cycle, and this might either prevent premature release of the primer before reaching the desired length, or to function as the first end of a "two-ended measuring tape" (see below), or perhaps both. Previous investigations have indicated the importance of the Pri2-CTD in RNA primer synthesis. For example, Pri2-CTD was shown to contribute to the simultaneous binding of the two first rNTPs by the primase Pri1 and to facilitate the dinucleotide base-pairing with the template DNA at the primer initiation site[29,44]. Consistent with this function, the recent Pol α–CST complex structure showed that the Pri2-CTD is located next to the Pri1 primase active site[36]. In our structure of the Pol α-T/P8 complex, the Pri2-CTD, located within the middle region of the Pri1 enzyme and in close proximity to its active site, facilitates the entry of incoming rNTP molecules into the Pri1 active site. While earlier studies showed that Pri2 can bind the 5′ end of the T/P[16], it was somewhat surprising to observe in our study that the Pri2-CTD remains bound to the 5′ ss/ds DNA of the T/P after the T/P handoff to the Pol1-core and during DNA primer extension (Fig. 5a). Therefore, Pri2-CTD binds the existing 5′ ss/ds DNA end while Pol1-core holds the growing 3′ end of the T/P duplex. Because both Pri2-CTD and the Pol1-core are linked to the Pol1-CTD–Pol12 platform by linker peptides, it is possible that both the Pri2-CTD and Pol1-core function as the two ends of a measuring tape that ultimately determines the final length of the RNA-DNA primer[28] (see step 6 in Fig. 6).

## Termination of primer synthesis

Our structural studies support a "linker stretching" hypothesis that determines the DNA nt counting mechanism of Pol α-primase that limits the primer length. In this view, the Pol1-core functions as one of the two ends of the measuring tape. We suggest that when it has extended the RNA primer by a full turn of the DNA double helix (i.e., 34 Å or ~10 base pairs), the Pol1-core fully stretches the 10-residue linker between Pol 1-core to the Pol1-CTD in the platform (see step 6 in Fig. 6), and that the Pri2-CTD that caps the other 5′ end of the T/P duplex may have also fully stretched the 20-residue linker to the Pri2-NTD. This proposed "linker stretching" hypothesis would then cause the release of Pol1-core from the T/P, leading to the termination of the RNA-DNA primer (see Fig. 6, step 6). This "linker stretching" hypothesis for how Pol α-primase counts the length of a primer is supported by recent mutagenesis studies of the linkers in Pol α, in which changing linker lengths correlated with changing of the primer lengths[28]. However, use of mutant enzyme may alter product lengths in unforeseeable ways. In fact, there is evidence that Pol α-primase counts primer length via an intrinsic decreased affinity for B-form DNA-DNA compared to A-form RNA-DNA, causing Pol α-primase to simply dissociate from the RNA-DNA hybrid primer after it extends DNA about 1 turn, to a distance that places the Pol1-core squarely onto duplex B-form DNA[24]. That study showed that Pol1-core alone extends a DNA primer with dNTPs to variable lengths that peak at 10–12 nt, but with a sizable portion reaching 20–30 nt, and this property is independent of the provided RNA primer size[24]. However, a recent study of Pol1-core showed a similar $K_d$ for binding to RNA-DNA as for binding of DNA-DNA which does not fully support a model that relies on dissociation of B form over A form duplex[27]. Thus, it is not yet firmly established how Pol α-primase counts nucleotides. In fact, the two hypotheses are not mutually exclusive and thus it is possible that both hypotheses, B-form DNA affinity and linker stretching, contribute to DNA synthesis termination. In fact, our structural study suggests a third possible mechanism, or contribution, to primer length measurement of the

RNA-DNA primer. Specifically, given the unexpected highly dynamic nature of this multiprotein enzyme it is possible that interdomain interactions among subunits may facilitate the dissociation of Pol1-core from DNA after the DNA reaches a certain length.

In summary, our systematic structural analysis has revealed the series of conformational changes Pol α-primase undergoes during primer synthesis. The structural changes suggest a very dynamic process with large conformational changes among the 4-subunit conserved complex that enable handoff of the RNA portion made by Pri1, to Pol1-core, in which all the conformers are linked by their common interactions with the Pol12-Pol1-CTD rigid platform that orchestrates the dance among the partners of the complex that determine the RNA-DNA finale through constraints imposed by the linkers to the primase and polymerase domains.

## Methods

### Proteins and nucleic acids

Yeast Pol α-primase was purified as previously described[45]. Proteins were dialyzed against 50 mM HEPES pH 7.5, 50 mM KGlu, 200 mM KAcetate, 1 mM DTT, 4 mM MgCl$_2$, aliquoted, snap frozen in liquid nitrogen, and stored at −80 °C. An SDS PAGE analysis of Pol α-primase used in this study is shown in Supplementary Fig. 1a. Equimolar oligonucleotide of template and primer were mixed in a PCR tube. The heat profile of the primer-template annealing was 95 °C for 2 min and 20 °C for 45 min. Prior to use, the product was stored at 4 °C. The oligonucleotide sequences used in this study are presented in Supplementary Table 2.

### In vitro assembly of various Pol α−T/P complexes

The 60-nt ssDNA template was mixed with Pol α-primase to introduce the primer initiation state. We used seven primers of different lengths (Supplementary Table 2): a 6-nt RNA primer (P6), a 7-nt RNA primer (P7), an 8-nt RNA primer (P8), a 10-nt RNA primer (P10), an 11-nt RNA primer (P11), and a 15-nt chimeric primer consisting of 10-nt RNA and 5-nt DNA(P15). These primers were individually annealed with the 60-nt DNA template to form the T/P substrate used to capture different priming states of the Pol α. To induce the RNA priming state, Pol α was incubated with the 6-nt, 7-nt, and 8-nt RNA primers annealed with the 60-nt DNA template (T/P6, T/P7, and T/P8), NTP, and 4% FA at 30 °C for 5 min before making cryo-EM grids. To capture the transition (RNA hand-off state from Pri1 to Pol1), Pol α was individually incubated with the 9-nt, 10-nt, and 11-nt RNA primers annealed with the 60-nt DNA template (T/P9, T/P10, and T/P11) at 30 °C for 5 min. To induce the DNA primer extension state, we incubated the enzyme with the 15-nt RNA-DNA chimeric primer annealed to the 60-nt DNA template (T/P15) using the same conditions as described above before making cryo-EM grids.

### Cryo-EM

We used a Vitrobot Mark IV (Thermo Fisher) to prepare cryo-EM grids of the various Pol α-primase samples. A 3-μl droplet of the samples at a final concentration of ~1 mg/ml was pipetted onto C-flat 2/1 holey carbon grids, treated by glow-discharge immediately before use. The grids were then incubated for 10 s at 6 °C in 100% humidity, blotted for 3 s, and then plunged into liquid ethane. EM grids were loaded into a Titian Krios electron microscope operated at 300 kV, and images were collected automatically in low-dose mode at a magnification of ×130,000 with an objective lens defocus ranging from −1.5 to −2.5 μm. A Gatan K3 Summit direct electron detector was used for image recording in the super-resolution mode with a pixel size of 0.828 Å at the sample level. The dose rate was 10 electrons per Å$^2$ per second and the total exposure time was 6 s. The total dose per micrograph was divided into a 30-frame movie, so each frame was exposed for 0.2 s. We used SerialEM[46] to collect one dataset of 10,519 raw movie micrographs for Pol α alone in the apo form, one dataset of 3357 raw movie

micrographs for Pol α-T complex, one dataset of 14,691 micrographs for the Pol α–T/P8 complex, one dataset of 13,433 micrographs for the Pol α–T/P10 complex, and one dataset of 17,525 micrographs for the Pol α–T/P15 complex.

## Image processing and 3D reconstruction

Relion-3.1 was used to carry out all image processing steps, including particle picking, 2D classification, 3D classification, 3D reconstruction and refinement, and postprocessing[47]. First, the individual movie frames of each micrograph were aligned and superimposed using the program Motioncorr2[48]. The contrast transfer function parameters of each aligned micrograph were calculated, and the CTF effect was corrected for each micrograph with the derived parameters in the CTFFIND4 program[49]. For each dataset, we manually picked about 2000 particles in different views to generate several 2D averages. And these averages were used as templates for subsequent automatic particle picking of the entire dataset.

For Pol α in the apo form, a total of 1,964,689 particles were initially picked (Supplementary Fig. 2). These raw particles were sorted according to the similarity to the 2D references; the bottom 10% of particles that had very low $z$-scores were deleted from the particle pool. 2D classification was then performed for the remaining particles, and particles belonging to the "bad" 2D classes (damaged or partial structure, or blurry averages with no structural details) were removed. In total, 799,378 good particles with complete Pol α-primase structural features were kept for the following 3D classification. We derived five 3D classes and one was chosen for further classification, leading to two 3D EM maps with an average resolution of 3.7 and 3.8 Å, respectively.

For the Pol α-T dataset (primer initiation state), 674,924 particles were initially picked (Supplementary Fig. 3). The dataset was processed according to the protocol used in the Pol α-primase apo form. After 2D classification, 490,120 particle images were retained for 3D classification. We generated five 3D classes, and the best class containing 36% of the particle images was selected for further analysis, leading to the final 3D map at an estimated resolution of 5.6 Å.

For the Pol α–T/P8 dataset (RNA synthesis state), a total of 2,791,251 particles were initially selected (Supplementary Fig. 4). The worst 10% of these that had very low $z$-scores were removed from the particle pool after being sorted based on similarity to the 2D reference. All remaining particles were subjected to a 2D classification, and those in poor classes were eliminated. A total of 1,459,235 good particles (with full Pol α-primase structural features) were retained for the subsequent 3D classification. One 3D class with a good Pri1-platform density was chosen for further data processing. We performed focused 3D classification by focusing on the Pol1-core–Pri2-CTD–T/P8 region. We obtained six 3D classes in this region leading to the final EM map, which had an estimated resolution of 4.8 Å.

For the Pol α–T/P10 dataset (post RNA hand-off state), a total of 2,532,900 particles were initially picked (Supplementary Fig. 5). Similar to the data processing of Pol α–T/P8, we performed 2D and 3D classifications to clean up the particle dataset. We chose two 3D classes with good densities in the Pri1-platform region and masked out the Pol1-core–Pri2-CTD–T/P10 region for focused 3D classification. 3D classes with good densities in the Pol-1-core–Pri2-CTD–T/P10 region were selected for further processing, resulting in the final EM map at an estimated average resolution of 4.5 Å.

For the Pol α–T/P15 dataset (DNA elongation state), a total of 3,787,982 particles were initially picked (Supplementary Fig. 6). After 2D classification, the bottom 10% of particles that had very low $z$-scores were deleted from the particle pool. This led to a selected dataset of 1,872,197 good particles. 3D classification of the selected dataset resulted in five 3D models, and the best one was selected for the final 3D reconstruction, leading to an EM map at the estimated overall resolution of 3.5 Å.

The reported resolutions for all EM maps were based on the Gold-standard Fourier shell correlation of the corresponding half maps at the cutoff threshold of 0.143. All EM maps were sharpened by using negative B-factors and compensated for the modulation transfer function of the detector. ResMap was used to assess the local resolution[50].

## Atomic modeling, refinement, and validation

The atomic structures of individual subunits of the *S. cerevisiae* Pol α-primase are available from previous crystallography and cryo-EM studies, including the yeast Pol1-core structure (4B08), the Pol12 structure (3FLO]), the Pri1 structure (4BPU), and the Pri2 structure (3LGB). Their models were extracted from the PDB coordinates and rigid-body fitted into the 3D EM maps in the COOT[51] and Chimera programs[52]. The DNA/RNA models were built based on the used primer sequences. The resulting full Pol α-primase models were refined by rigid body refinement of individual chains in the PHENIX program[53], and subsequently adjusted manually in COOT guided by residues with bulky side chains like Arg, Phe, Tyr, and Trp. The models were then refined in real space by phenix.real_space_refine and in reciprocal space by PHENIX with the application of secondary structure and stereochemical constraints. The structure factors (including phases) were calculated by Fourier transform of the experimental EM map with the program Phenix.map_to_structure_factors. The final models were validated using MolProbity[54]. Structural figures were prepared in Chimera and Pymol (https://www.pymol.org). The model statistics are listed in Supplementary Table 1, and the correlation between models and maps is listed in Supplementary Table 3.

## Reporting summary

Further information on research design is available in the Nature Portfolio Reporting Summary linked to this article.

## Data availability

The 3D EM maps of the *S. cerevisiae* apo Pol α in states I at 3.7 Å and II at 3.8 Å, Pol α bound to template ssDNA (state III) at 5.6 Å, Pol α with T/P8 at 4.8 Å, Pol α with T/P10 at 4.5 Å, and Pol α with T/P15 at 3.5 Å resolution have been deposited in the EMDB under accession codes EMD-29345, EMD-29346, EMD-29347, EMD-29349, EMD-29351, and EMD-29352 respectively. The corresponding atomic models have been deposited in the Protein Data Bank under accession codes 8FOC, 8FOD, 8FOE, 8FOH, 8FOJ, and 8FOK, respectively. Source data are provided with this paper.

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

## Acknowledgements

Cryo-EM data were collected at the David Van Andel Advanced Cryo-Electron Microscopy Suite at the Van Andel Institute. We thank G. Zhao and X. Meng (VAI) for their help with data collection, and Olga Yurieva (RU) for Pol α purification. This study was supported by the US National Institutes of Health grants GM131754 (to H.L.) and GM115809 (to M.E.O.), the Howard Hughes Medical Institute (to M.E.O.), and the Van Andel Institute (to H.L.).

## Author contributions

Z.Y., R.G., H.L., and M.E.O. conceived and designed experiments. R.G. conditioned the Pol α prep for application to EM grids. Z.Y. performed the EM experiments, image processing, 3D reconstruction, and atomic modeling. Z.Y., H.L., and M.E.O. analyzed the data and wrote the manuscript.

## Competing interests

The authors declare no competing interests.
