## [Peer Review File · Nature Communications]

Molecular choreography of primer synthesis by the eukaryotic Pol α -primaseREVIEWER COMMENTS

Reviewer #1 (Remarks to the Author):

In this paper, Yuan and colleagues investigate the structural mechanism of the interplay of the 4 subunits of yeast Polymerase α /primase in primer synthesis by the primase, primer handoff from the primase to the polymerase, primer extension by the polymerase and substrate release. This mechanism is of critical importance in eukaryotic DNA replication, and it has remained rather obscure due to the high flexibility of this 4-subunit enzyme, which prevented a detailed structural characterization. The authors here report 5 different cryo-EM structures, ranging from 3.5 to 5.6Å resolution, of Pol α /primase in apo form (resting state), bound to a ssDNA template (preinitiation state), bound to a 8 bp RNA/DNA substrate (RNA synthesis state), bound to a 10 bp RNA/DNA substrate (RNA handoff state), and bound to a 15 bp RNA/DNA substrate (elongation state). Based on the analysis of the rearrangements of the Pol α subunits (Pol1, Pol12, Pri1, and Pri2) and substrate interactions in these structures, the author propose a mechanism for primer synthesis and release. In this model, the primase and polymerase enzymatic units (Pri1 and Pol1-core) are flexibly tethered to a stable platform formed by Pol1-CTD and Pol12, and their mutual position is dictated by the type of RNA/DNA substrate bound to the complex. Based on structural constraints imposed to the mobile Pol α subunits by the length of the RNA primer, the author propose that Pri1 can synthesize a maximum of 10 ribonucleotides before the RNA/DNA substrate is handed off to Pol1 for further extension, and that Pol1 may add a maximum of 10 deoxynucleotides before the substrate is released from the complex for another cycle of primer synthesis. These substrate lengths appear in agreement with previous biochemical observations. The structural constraints governing product release are the stable attachment of the 5' end of the primer strand to Pri2-NTD and the maximum extension of the flexible linkers connecting Pol1-core to Pol1-CTD and Pri2-NTD to Pri2-CTD.

Overall, this study addresses a very important and difficult problem. Many observations are novel, and the mechanism presented for Pol α function is attractive. I have, however, concerns about the quality of the models that have been built into the cryo-EM maps. Because of the rather poor map-to-model correlation for important parts of the models (in general, the Q-scores in the validation reports are significantly lower than those expected from maps at the reported resolutions (Pintilie, Nature Methods, 2020)), the conclusions drawn about the mechanism remain vague. In further submissions, the authors should in my view include per-chain map-to-model correlation coefficients, like those output by Phenix for instance (Afonine, Acta Crystallogr D Struct Biol., 2018), and map-to-model FSC resolution cutoffs in the data refinements/statistics table. Model portions with unreliably low map-to-model correlation due to conformational flexibility should not be included in the deposited model.

The authors should provide convincing evidence for the reliable modelling of the critical interfaces between Pol α subunits in all 5 structures presented, with a particular attention to those that cannot be directly compared to previously published high-resolution structures of Pol α in analogous states and in absence of additional factors (i.e., Pol α in preinitiation state, in RNA synthesis state, in RNA handoff state, and in elongation state). Importantly, better evidence should be provided for the quality of the modelling of the RNA/DNA substrates, and the important interactions of the substrates with the different Pol α subunits.

Pol α in the primer initiation state

The authors state that the resolution achieved for this map (5.6Å) is not sufficient to resolve the bound ssDNA. Therefore, the ssDNA bound to human Pol α -primase (Xe et al, Nature, 2022; PDB: 8D0K) was used as a placeholder for the template DNA in the yeast structure (Supp Fig. 8). However, the conformations of human and yeast pol alpha in the two structures are radically different (Supp Fig. 8), supposedly because in the human structure Pol α is bound to CST. Consequently, it appears difficult to extrapolate the position of ssDNA in the yeast structure from the human structure. As the DNA is not resolved in the yeast structure, is there any additional evidence that DNA is bound to Pol α ? Is DNA bound to the active site of Pri1?

A possibility to enhance the DNA features in this map is to carry out multi-body refinement (Nakane, Elife, 2018) splitting Pol α into discrete bodies according to their relative mobility (and including DNA in the Pri1 body). Multi-body refinement may also be attempted to improve the features of the structures evaluated hereafter.

Pol α in the RNA synthesis state

In this map (reported at 4.9Å resolution), an RNA/DNA substrate was modelled; the 5' end of the primer appears to interact with Pri2-CTD, while the 3' end approximates the active site of Pri1. These interactions are critical for the discussion and should be presented in a more detailed way. Cryo-EM density and corresponding models for interacting nucleotides and protein residues should be displayed. According to the validation reports, fitting of RNA/DNA models is rather poor in this structure (67% of nucleotides of the template strand have <40% all-atom inclusion in the map). Regarding the interaction of Pol1-core with RNA/DNA, the authors write (pg. 5): "the thumb is stabilized by the T/P8 duplex region and becomes visible in the post RNA hand-off state indicating an incremental formation of the active DNA polymerase pocket". A picture showing this interface including the interacting residues should be shown, together with the corresponding portion of the cryo-EM map.

Pol α in the RNA post-handoff state

In this 4.8Å resolution map, a 10 bp T/P template was modelled. The validation report shows very low inclusion of the modelled nucleotides (all-atom inclusion < 40% for 82% and 100% of residues in the template and primer strands, respectively). Because the authors make assumptions about the protein/substrate interactions in this model (for instance (pg. 5): "The post RNA hand-off state structure shows that the primer 3' end of the T/P10 is engaged with the DNA Pol1 active site"), they should provide better evidence of the model reliability.

Minor comments

- Reference 34 is quoted as reporting human pol α structures, however that paper reports tetrahymena pol α structures.
- On page 4, text points to Fig. S8c-e, but these panels are absent from the figure
- On page 5, labels for quoted Fig. 3d-e are missing in the figure

Reviewer #2 (Remarks to the Author):

The DNA synthesis by the leading and lagging strand DNA polymerases requires a primer annealed to the DNA template. In eukaryotes, the primer is an RNA-DNA hybrid and synthesized by polymerase α (Pol α). Pol α is a heterotetrametric complex containing the catalytic primase subunit Pri1, the catalytic DNA polymerase subunit Pol1, and two regulatory subunits Pri2 and Pol12. Despite decades of efforts, mechanisms of Pol α in synthesizing an RNA-DNA hybrid primer with a limited length are still not fully understood at molecular level. In this manuscript, Yuan et al. determined the structures of a set of key intermediates of yeast Pol α throughout the reaction cycle of primer synthesis, by incubating Pol α with different template/primer (T/P) DNA substrates. The structural comparisons reveal large conformational changes between these functional intermediates of Pol α , providing plenty of structural findings for answering several key questions related to the primer synthesis, such as the substrate hand-off from primase to polymerase and the control of the primer length. This work has made significant contributions to the field of DNA replication.

Major concerns:

1. Many structural descriptions in the main text were not clearly presented in the figures, or lacked supporting figure, making the manuscript hard to follow. These issues should be addressed throughout the manuscript.

Some examples are as follows:

(1) Page 4 and paragraph 2, "such that the OB domain of Pol12 blocks the Pol1-core from binding a polynucleotide substrate from the top, and the Pri2-CTD binds the Pol1-core thumb and palm subdomains from the left side, also preventing access of a DNA template to the Pol1-core catalytic site (Fig. 2a, c)." The DNA binding inhibitions by both Pol12 and Pri2-CTD could not be concluded from these two panels. OB, thumb and palm were not labeled.

(2) Page 4 and paragraph 3, "However, the overall structure of the primer initiation state yeast Pol α -primase is similar to a recently reported structure of the human Pol α -T (ssDNA) preinitiation complex stabilized by the telomeric CST complex such that the template DNA was resolved (Fig. S8c-e)". The structural similarity between primer initiation state of yeast Pol α -primase and human Pol α -T (ssDNA) preinitiation complex could not be seen in Fig. S8. On the contrary, Fig. S8b showed a large conformational difference between these two structures.

(3) Page 4 and paragraph 3, "Therefore, the ssDNA bound to human Pol α -primase is used as a placeholder for the template DNA in the yeast Pol α -primase (Fig. 2b)". DNA is not labeled or indicated in Fig. 2.

(4) Page 4 and paragraph 4, "Therefore, both Pri1 and Pol1-core engage the template DNA, but their catalytic sites are $\sim 30 \text{ \AA}$ – equivalent to 10 nt – apart from each other (Fig. 2b, 2d)." The position of DNA should be indicated, and the distance of 30 \AA should be labeled.

(5) Page 4 and paragraph 5, "In this RNA synthesizing and pre hand-off RNA state, the Pri1 and the platform (Pol1-CTD– Pol12) are configured like in the template-bound initiation state, suggesting that Pri1 maintains the same pose during synthesis of the first 8 nt of the RNA primer." This sentence lacked supporting figure. As shown in Fig. 1d-e, Pri1 showed significant structural difference between the structures in the primer initiation and RNA synthesis state. Please clarify.

(6) Page 5 and paragraph 1, "transition to the RNA synthesis state (after synthesizing an 8-nt RNA primer) involves the Pri2-CTD turning and moving downwards by 30 \AA to accommodate the 8-bp RNA/DNA duplex. The Pri2-CTD binding site to the 5'-end of the duplex is approximately 40 \AA from the Pri1 catalytic site." These two sentences required supporting figures and the distances should be labeled in the figures.

"This distance is equivalent to the length of one full-turn of A-form RNA/DNA duplex that is the average RNA primer length." should provide a reference.

(7) Page 5 and paragraph 2, "We note that the thumb of Pol1-core is flexible and invisible in the initiation state of both the human Pol α -ssDNA and the yeast Pol α -T, but the thumb is stabilized by the T/P8 duplex region and becomes visible in the post RNA hand-off state indicating an incremental formation of the active DNA polymerase pocket." A supporting figure is required. The implications of "is stabilized" and "becomes visible" should also be clearly distinguished by the figure presentation.

(8) Page 5 and paragraph 4, "In our full-length Pol α -primase structure, the thumb binds P2-5, and the palm binds P5-10 (Fig. 4a)." Since the specific nucleotide numbers were mentioned in this sentence, a figure panel showing the interaction details between thumb/palm and P2-5/P5-10 with the interacting side chains shown and labeled, should be added.

(9) Page 5 and paragraph 4, "Interestingly, during the large-scale transition of Pol1-core, the N-terminus of Pol1-core "pushes" on the Pri1-NTD to release Pri1 from the template DNA. This transient interaction between Pol1-core and Pri1-NTD". Please provide a figure for this sentence. Moreover, what does the "transient interaction" mean exactly?

(10) Page 7 and paragraph 2, the distances between the primase and polymerase for most of the conformational states (except the one of 70-\AA distance in the apo state) were not shown in figures or described in Result. Please describe the method and criterion for these distance measurements and label these distances in relevant figures.

2. In several figures, the atomic model of the primer initiation state was used in the structural comparison with the apo state, the RNA synthesis state and the structure of human primer initiation state. Considering the relative low resolution of the primer initiation state (5.6 \AA), the model-map fitting figures should also be provided in Fig. S4 to confirm the accuracy of the atomic model.

3. Related to the explanation for the observation that only 10-nt RNA primer can trigger the primer hand-off (Page 5, paragraph 4, "This T/P binding mode may explain why a shorter RNA primer (6-9

nt) does not trigger the primer hand-off: a shorter primer can stably bind the thumb but not the palm, and a stable primer binding by the palm perhaps requires P5-10. Therefore, only when the RNA primer reaches 10 nt can the Pol1-core catalytic pocket compete with the Pri1 and stably engage the T/P10 and be converted to the polymerase pose"). The tight interaction between Pol1-core and primer is more likely a result of the primer hand-off than the cause. This description should be weakened and the related discussion in Page 7 paragraph 3 should be largely modified, otherwise, additional evidences are required.

Minor comments:

1. Many figure citations were incorrect or inaccurate, which should be revised.

Just to name one: Page 4, paragraph 3, "Upon binding template ssDNA, the Pol1-core dissociates from the platform, leading to a drastically different Pol α -primase structure to form the template-bound primer initiation state (Fig. 2b-d, Fig. S3, S8b, Supplementary video 1)." Fig. 2b-d should be Fig. 2d-f. The authors should carefully go through the manuscript to make corrections.

2. Page 5 and paragraph 4, "The dumbbell then pivots on the Pri2-CTD and rotates counterclockwise $\sim 40^\circ$ away from the Pol1-core (Fig. 4c)". The 40° -rotation angle should be labeled in the figure panel.

3. Page 6 and paragraph 4, "We have visualized all major intermediate states of a eukaryotic Pol α -primase in this study". "All" is not appropriate.

4. Supporting figures are not cited in order.

Reviewer #3 (Remarks to the Author):

The manuscript by Li and co-workers presents an impressive structural evaluation of the dynamic activities of yeast DNA Pol alpha by cryoelectron microscopy. The interplay between the subunits has been a subject of interest by many groups since the discovery of a DNA primase activity associated with DNA polymerase alpha in 1982, and this work demonstrates the elaborate orchestration of numerous steps in the process of primer formation followed by DNA polymerization. The manuscript is very well written and interesting, and the figures and video illustrate very effectively the structural findings. The cartoons are especially helpful to visualize the enzymatic process. Though it is difficult for a non-expert in the cryoelectron microscopy field to evaluate comprehensively the structural data collected, the description of the methodology and the data shown in the supplemental figures is thorough and compelling. This reviewer would have appreciated the inclusion of numerous panels of the reconstructed images presented in the main figures alongside the structures derived from the data. At present, only figure 2e shows such a reconstructed micrograph and it is really too small to see, and should be presented on the scale of the structures.

POINT-BY-POINT RESPONSES TO REVIEWER COMMENTS

Reviewer #1 (Remarks to the Author):

In this paper, Yuan and colleagues investigate the structural mechanism of the interplay of the 4 subunits of yeast Polymerase α /primase in primer synthesis by the primase, primer handoff from the primase to the polymerase, primer extension by the polymerase and substrate release. This mechanism is of critical importance in eukaryotic DNA replication, and it has remained rather obscure due to the high flexibility of this 4-subunit enzyme, which prevented a detailed structural characterization. The authors here report 5 different cryo-EM structures, ranging from 3.5 to 5.6Å resolution, of Pol α /primase in apo form (resting state), bound to a ssDNA template (preinitiation state), bound to a 8 bp RNA/DNA substrate (RNA synthesis state), bound to a 10 bp RNA/DNA substrate (RNA handoff state), and bound to a 15 bp RNA/DNA substrate (elongation state). Based on the analysis of the rearrangements of the Pol α subunits (Pol1, Pol12, Pri1, and Pri2) and substrate interactions in these structures, the author propose a mechanism for primer synthesis and release. In this model, the primase and polymerase enzymatic units (Pri1 and Pol1-core) are flexibly tethered to a stable platform formed by Pol1-CTD and Pol12, and their mutual position is dictated by the type of RNA/DNA substrate bound to the complex. Based on structural constraints imposed to the mobile Pol α subunits by the length of the RNA primer, the author propose that Pri1 can synthesize a maximum of 10 ribonucleotides before the RNA/DNA substrate is handed off to Pol1 for further extension, and that Pol1 may add a maximum of 10 deoxynucleotides before the substrate is released from the complex for another cycle of primer synthesis. These substrate lengths appear in agreement with previous biochemical observations. The structural constraints governing product release are the stable attachment of the 5' end of the primer strand to Pri2-NTD and the maximum extension of the flexible linkers connecting Pol1-core to Pol1-CTD and Pri2-NTD to Pri2-CTD.

Overall, this study addresses a very important and difficult problem. Many observations are novel, and the mechanism presented for Pol α function is attractive. I have, however, concerns about the quality of the models that have been built into the cryo-EM maps. Because of the rather poor map-to-model correlation for important parts of the models (in general, the Q-scores in the validation reports are significantly lower than those expected from maps at the reported resolutions (Pintilie, Nature Methods, 2020), the conclusions drawn about the mechanism remain vague. In further submissions, the authors should in my view include per-chain map-to-model correlation coefficients, like those output by Phenix for instance (Afonine, Acta Crystallogr D Struct Biol., 2018), and map-to-model FSC resolution cutoffs in the data refinements/statistics table. Model portions with unreliably low map-to-model correlation due to conformational flexibility should not be included in the deposited model.

The authors should provide convincing evidence for the reliable modelling of the critical interfaces between Pol α subunits in all 5 structures presented, with a particular attention to those that cannot be directly compared to previously published high-resolution structures of Pol α in analogous states and in absence of additional factors (i.e., Pol α in preinitiation state, in RNA synthesis state, in RNA handoff state, and in elongation state). Importantly, better evidence should be provided for the quality of the modelling of the RNA/DNA substrates, and the important interactions of the substrates with the different Pol α subunits.

We thank the reviewer for the constructive criticism and valuable feedback. We have carefully examined our models and maps and identify three causes for the poor fitting: (i) Multiple flexible regions in PolA caused the EM density of Pri1 or Pol1 to appear fragmented when they take on different positions around Pri2, particularly when they are not actively synthesizing an RNA primer or DNA primer. Although docking of the known structures into the EM map was confident, the weak densities affected the correlation between model and map. (ii) We did not verify the model-to-map fitting before deposition. The rigid body model was slightly misaligned with the map when it was inspected and saved in UCSF Chimera. (iii) The chosen contour level of the EM map during deposition can influence the model-to-map fitting. The inadvertent use of a bit too high contour level had resulted in the poor fitting, as the EM density distribution was uneven.

To address the reviewer's concerns, we have made the following improvements in revision: (1) We have performed additional 3D classification, multibody refinement, and 3D-flex analysis. These have led to appreciable improvement in the partially flexible and weak-density regions. (2) We have improved all six Q-scores to the expected average Q-scores (See attached panel below, calculated using $y = -0.178x + 1.119$ as in the Nature Methods paper). (3) We now provide the model-vs-map and model-vs-half map FSCs in the supplemental figures for all six PolA states captured in this study (**Supplemental Figures 2-6**). (4) We now present the per-chain correlations for all six states in **Supplemental Table 3**, as suggested by the reviewer. (5) We now include additional Supplemental figures to display model-map superposition, emphasizing the subunit-subunit and subunit-substrate interactions (**Supplemental Figures 3f, 4f, 5f, 6e-g**). (6) We have redeposited six maps and associated models in the PDB database and obtained new validation reports.

Pol α in the primer initiation state

The authors state that the resolution achieved for this map (5.6Å) is not sufficient to resolve the bound ssDNA. Therefore, the ssDNA bound to human Pol α-primase (Xe et al, Nature, 2022; PDB: 8D0K) was used as a placeholder for the template DNA in the yeast structure (Supp Fig. 8). However, the conformations of human and yeast pol alpha in the two structures are radically different (Supp Fig. 8), supposedly because in the human structure Pol α is bound to CST. Consequently, it appears difficult to extrapolate the position of ssDNA in the yeast structure from the human structure. As the DNA is not resolved in the yeast structure, is there any additional evidence that DNA is bound to Pol α? Is DNA bound to the active site of Pri1? A possibility to enhance the DNA features in this map is to carry out multi-body refinement (Nakane, Elife, 2018) splitting Pol α into discrete bodies according to their relative mobility (and including DNA in the Pri1 body). Multi-body refinement may also be attempted to improve the features of the structures evaluated hereafter.

Our cryo-EM analysis of Pol α with and without ssDNA template showed that Pol α undergoes a significant conformational change in the presence of ssDNA template. Specifically, comparison of 2D averages of Pol α with Pol α-ssDNA clearly demonstrates the release of Pol1 from the Pol12-Pri1 platform, as depicted in **Supplemental Figure 8a**. This observation indicates that template binding does activate Pol α in vitro. Following the reviewer's suggestion, we have used Relion multibody refinement and CryoSPARC 3D-flex to reanalyze the highly flexible Pri2 and Pol1. These refinements somewhat improved the reported resolutions, yet the density near Pri2-CTD still lacks definition to be assigned to the DNA template with confidence. Therefore, the ssDNA template is not included in our revised model.

Pol α in the RNA synthesis state. In this map (reported at 4.9 Å resolution), an RNA/DNA substrate was modelled; the 5' end of the primer appears to interact with Pri2-CTD, while the 3' end approximates the active site of Pri1. These interactions are critical for the discussion and should be presented in a more detailed way. Cryo-EM density and corresponding models for interacting nucleotides and protein

residues should be displayed. According to the validation reports, fitting of RNA/DNA models is rather poor in this structure (67% of nucleotides of the template strand have <40% all-atom inclusion in the map). Regarding the interaction of Pol1-core with RNA/DNA, the authors write (pg. 5): “the thumb is stabilized by the T/P8 duplex region and becomes visible in the post RNA hand-off state indicating an incremental formation of the active DNA polymerase pocket”. A picture showing this interface including the interacting residues should be shown, together with the corresponding portion of the cryo-EM map.

Thank you for the valuable suggestion. As mentioned earlier, we have remodeled the structure, resulting in a significant improvement in the model-to-map fitting. We have now included new panels to illustrate the correspondence between the model and the map (**Supplemental Figure 4f**). We have also included a new figure to better visualize the interaction between the thumb and T/P8 (**Supplemental Figure 10a-b**).

Pol α in the RNA post-handoff state. In this 4.8 Å resolution map, a 10 bp T/P template was modelled. The validation report shows very low inclusion of the modelled nucleotides (all-atom inclusion < 40% for 82% and 100% of residues in the template and primer strands, respectively). Because the authors make assumptions about the protein/substrate interactions in this model (for instance (pg. 5): “The post RNA hand-off state structure shows that the primer 3' end of the T/P10 is engaged with the DNA Pol1 active site”), they should provide better evidence of the model reliability.

We have corrected the model-map misalignment, resulting in significantly improved atom inclusion and Qscore. Please see revised validation reports. We have now included revised model-vs-map FSCs (**Supplemental Figure 5c**) and the new per-chain CC (**Supplemental Table 3**) that show improved alignment and the model reliability. In revision, we have added new panels to show model-to-map superimposition as well as detailed substrate-protein interactions (**Supplemental Figure 5f**).

Minor comments

-Reference 34 is quoted as reporting human pol α structures, however that paper reports tetrahymena pol α structures.

Thanks for catching this error. We have revised accordingly.

- On page 4, text points to Fig. S8c-e, but these panels are absent from the figure
We have corrected this (see revised **Supplemental Figure 8b,c,d,e**)

- On page 5, labels for quoted Fig. 3d-e are missing in the figure
We appreciate this comment and have revised **Fig. 3** accordingly.

Reviewer #2 (Remarks to the Author):

The DNA synthesis by the leading and lagging strand DNA polymerases requires a primer annealed to the DNA template. In eukaryotes, the primer is an RNA-DNA hybrid and synthesized by polymerase α (Pol α). Pol α is a heterotetrametric complex containing the catalytic primase subunit Pri1, the catalytic DNA polymerase subunit Pol1, and two regulatory subunits Pri2 and Pol12. Despite decades of efforts, mechanisms of Pol α in synthesizing an RNA-DNA hybrid primer with a limited length are still not fully understood at molecular level. In this manuscript, Yuan et al. determined the structures of a set of key intermediates of yeast Pol α throughout the reaction cycle of primer synthesis, by incubating Pol α with different template/primer (T/P) DNA substrates. The structural comparisons reveal large conformational changes between these functional intermediates of Pol α , providing plenty of structural findings for answering several key questions related to the primer synthesis, such as the substrate hand-off from primase to polymerase and the control of the primer length. This work has made significant contributions to the field of DNA replication.

Major concerns:

1. Many structural descriptions in the main text were not clearly presented in the figures, or lacked

supporting figure, making the manuscript hard to follow. These issues should be addressed throughout the manuscript. Some examples are as follows: (1) Page 4 and paragraph 2, “such that the OB domain of Pol12 blocks the Pol1-core from binding a polynucleotide substrate from the top, and the Pri2-CTD binds the Pol1-core thumb and palm subdomains from the left side, also preventing access of a DNA template to the Pol1-core catalytic site (Fig. 2a, c).” The DNA binding inhibitions by both Pol12 and Pri2-CTD could not be concluded from these two panels. OB, thumb and palm were not labeled.

Thanks for the valuable feedback. We have now labeled the OB, thumb, and palm domains in revised **Figure 2a-f** for clarity, have revised many of the other figures, and have added labels in several structures presented throughout the manuscript.

(2) Page 4 and paragraph 3, “However, the overall structure of the primer initiation state yeast Pol α -primase is similar to a recently reported structure of the human Pol α -T (ssDNA) preinitiation complex stabilized by the telomeric CST complex such that the template DNA was resolved (Fig. S8c-e)”. The structural similarity between primer initiation state of yeast Pol α -primase and human Pol α -T (ssDNA) preinitiation complex could not be seen in Fig. S8. On the contrary, Fig. S8b showed a large conformational difference between these two structures.

We agree with the reviewer and acknowledge that there are appreciable differences that become more apparent upon alignment. However, the overall architecture is similar in that the Pol12-Pri2NTD-Pri1 platform is located on one side, the Pol1-NTD is situated on the other side, and with the Pri2-CTD holding the ssDNA in the middle. To address the reviewer’s concern, we have revised the text by noting that “the detailed structures differ”.

(3) Page 4 and paragraph 3, “Therefore, the ssDNA bound to human Pol α -primase is used as a placeholder for the template DNA in the yeast Pol α -primase (Fig. 2b)”. DNA is not labeled or indicated in Fig. 2.

We have modified **Fig. 2d** and added **Supplemental Figure 8c** to show approximate template DNA position. The revised sentence reads: “Therefore, the ssDNA bound to human Pol α -primase is used as a placeholder for the template DNA in the yeast Pol α -primase (see **Fig. 2d** and **Fig. S8c**).”

(4) Page 4 and paragraph 4, “Therefore, both Pri1 and Pol1-core engage the template DNA, but their catalytic sites are ~ 30 Å – equivalent to 10 nt – apart from each other (Fig. 2b, 2d).” The position of DNA should be indicated, and the distance of 30 Å should be labeled.

We have labelled the distance in revised **Fig. 2d** and shown the distance between Pri1 and Pol1 in all states in **Supplemental Figure 9a**. Accordingly, we have revised the sentence in the main text to read: “Therefore, both Pri1 and Pol1-core engage the template DNA, but their catalytic sites are ~ 49.5 Å apart (**Fig. S9**).”.

(5) Page 4 and paragraph 5, “In this RNA synthesizing and pre hand-off RNA state, the Pri1 and the platform (Pol1-CTD– Pol12) are configured like in the template-bound initiation state, suggesting that Pri1 maintains the same pose during synthesis of the first 8 nt of the RNA primer.” This sentence lacked supporting figure. As shown in Fig. 1d-e, Pri1 showed significant structural difference between the structures in the primer initiation and RNA synthesis state. Please clarify.

In the revised manuscript, we have added **Supplemental Figure 8c-e** to compare the template-bound initiation state with the RNA synthesizing state and acknowledged that all subunits undergo some degree of conformational changes, although the basic architecture of the Pol1-CTD-Pol12-Pri1 axis is similar between the template binding and RNA synthesis states. To simplify the presentation, we now focus on the three regions directly related to RNA synthesis: Pri1, Pol1-core, and Pri2-CTD; they are aligned using Pri1 as a reference in revised Fig. 3 (i.e., panels **Fig. 3c-f**)).

The revised text reads:

“During RNA synthesis, the overall architecture is configured similar to the initiation state with bound template. The Pri1 and platform (Pol1-CTD–Pol12) form an axis on one side, while Pol1-core, Pri2-CTD, and template DNA swing around this axis on the other side (**Figs. S8c-e**). Although all subunits undergo conformational changes during the synthesis of the first 8 nt of the RNA primer, for simplicity, we will focus only on the three subunits directly involved in RNA synthesis and use Pri1 as the reference to align these three subunits (**Fig. 3c-f**).”.

(6) Page 5 and paragraph 1, “transition to the RNA synthesis state (after synthesizing an 8-nt RNA primer) involves the Pri2-CTD turning and moving downwards by 30 Å to accommodate the 8-bp RNA/DNA duplex. The Pri2-CTD binding site to the 5'-end of the duplex is approximately 40 Å from the Pri1 catalytic site.” These two sentences required supporting figures and the distances should be labeled in the figures. “This distance is equivalent to the length of one full-turn of A-form RNA/DNA duplex that is the average RNA primer length.” should provide a reference.

We have added **Supplemental Figure 10a-b** which shows two structures that point out the 30 Å and 40 Å distances in the sentences that the reviewer refers to, and we have also included two references for the average RNA primer length as requested.

The revised text reads:

“...transition to the RNA synthesis state (after synthesizing an 8-nt RNA primer) involves the Pri2-CTD turning and moving downwards by 30 Å to accommodate the 8-bp RNA/DNA duplex (**Fig. S10a**). The Pri2-CTD binding site to the 5'-end of the duplex is approximately 40 Å from the Pri1 catalytic site (**Fig. S10b**). This distance is equivalent to the length of one full-turn of A-form RNA/DNA duplex that is the average RNA primer length^{23,36}.”

(7) Page 5 and paragraph 2, “We note that the thumb of Pol1-core is flexible and invisible in the initiation state of both the human Pol α -ssDNA and the yeast Pol α -T, but the thumb is stabilized by the T/P8 duplex region and becomes visible in the post RNA hand-off state indicating an incremental formation of the active DNA polymerase pocket.” A supporting figure is required. The implications of “is stabilized” and “becomes visible” should also be clearly distinguished by the figure presentation.

Thanks for this suggestion. We have now added **Supplemental Figure 10a** to show the thumb tip, and the revised text reads:

“We note that the thumb of Pol1-core is flexible and not resolved in the initiation state of both the human Pol α -ssDNA and the yeast Pol α -T (**Fig. S10a**), but the thumb is clearly observed and thus perhaps stabilized by the T/P8 duplex region in the post RNA hand-off state indicating an incremental formation of the active DNA polymerase pocket in this intermediate (**Fig. S10a**).”

(8) Page 5 and paragraph 4, “In our full-length Pol α -primase structure, the thumb binds P2-5, and the palm binds P5-10 (Fig. 4a).” Since the specific nucleotide numbers were mentioned in this sentence, a figure panel showing the interaction details between thumb/palm and P2-5/P5-10 with the interacting side chains shown and labeled, should be added.

We thank the reviewer for this suggestion. We have revised the **Fig. 4a** bottom panel to provide the reader with more specific information on the binding details between the primer and Pol.

(9) Page 5 and paragraph 4, “Interestingly, during the large-scale transition of Pol1-core, the N-terminus of Pol1-core “pushes” on the Pri1-NTD to release Pri1 from the template DNA. This transient interaction between Pol1-core and Pri1-NTD”. Please provide a figure for this sentence. Moreover, what does the “transient interaction” mean exactly?

Thanks for this query. We do feel that **Fig. 4b-d** already describes the process of conformation changes, but we agree that further clarification is needed, and especially for the use of the term “transient interaction”.

We have revised the text as follows:

“In the PolA apo state, we observed that Pol1-NTD interacts with Pri1-NTD in ~50% of the particle population, yet the Pri1 EM density is weak and broken. This indicates to us a “transient” nature of the interaction. Transitioning from PolA-T/P8 to PolA-T/P10, we observed that Pol1-NTD moves towards Pri1-NTD to bind the T/P10 duplex (**Fig. 4b-d**). Although the transition step was not captured, we hypothesize that Pol1-NTD interacts with Pri1-NTD during this transition, leading to the release of Pri1 from template DNA. We propose that during the large-scale transition of Pol1-core, the N-terminus of Pol1-core appears to “push” on the Pri1-NTD to release Pri1 from the template DNA (**Fig. 4b-d**). This presumed “transient interaction” between Pol1-core and Pri1-NTD was also observed in the apo enzyme conformer II structure (**Fig. 4b** lower right panel). In the completed post RNA hand-off to Pol1 state, Pol1-core binds the T/P10 in a catalytic pose but does not interact with other protein components, perhaps liberating the DNA polymerase module for primer extension (**Fig. 4b-d**); the released Pri1 is connected to the Pol1-CTD–Pol12 platform via Pri2-NTD, and they form a dumbbell-like structure. The dumbbell then pivots on the Pri2-CTD and rotates counterclockwise ~40° away from the Pol1-core (**Fig. 4c-d**). Thus, in the “post RNA hand-off to Pol1” state, the Pol1-CTD–Pol12 platform becomes closer to the Pol1-core, perhaps enabling the Pol1-core to bind to the Pol1-CTD in the next stage. But the current pose is compatible with DNA synthesis.”

(10) Page 7 and paragraph 2, the distances between the primase and polymerase for most of the conformational states (except the one of 70-Å distance in the apo state) were not shown in figures or described in Result. Please describe the method and criterion for these distance measurements and label these distances in relevant figures.

We have now revised the manuscript to illustrate and explain the distance measurements. We have included a new **Supplemental Figure 9a-b** to illustrate the distance measurements, and the explanation of the distances is described in the legend to **Supplemental Figure 9**.

The explanation of distance measurements reads: “The distance is measured between two positively charged amino acids in the catalytic pockets of Pri1 and Pol1, *between Pri1 K326 and Pol1 R917*, respectively.” Because the main figures are already extensive, we show this information in the supplemental figure.

2. In several figures, the atomic model of the primer initiation state was used in the structural comparison with the apo state, the RNA synthesis state and the structure of human primer initiation state. Considering the relative low resolution of the primer initiation state (5.6 Å), the model-map fitting figures should also be provided in Fig. S4 to confirm the accuracy of the atomic model.

Thanks for the suggestion and we fully agree. We have now added model-vs-map for all the six maps described in the manuscript in **Supplemental Figures 2-6**.

3. Related to the explanation for the observation that only 10-nt RNA primer can trigger the primer hand-off (Page 5, paragraph 4, “This T/P binding mode may explain why a shorter RNA primer (6-9 nt) does not trigger the primer hand-off: a shorter primer can stably bind the thumb but not the palm, and a stable primer binding by the palm perhaps requires P5-10. Therefore, only when the RNA primer reaches 10 nt can the Pol1-core catalytic pocket compete with the Pri1 and stably engage the T/P10 and be converted to the polymerase pose”). The tight interaction between Pol1-core and primer is more likely a result of the primer hand-off than the cause. This description should be weakened and the related discussion in Page 7 paragraph 3 should be largely modified, otherwise, additional evidences are required.

We agree and have revised accordingly:

“We therefore speculate that only when the RNA primer reaches 10 nt can the Pol1-core catalytic pocket compete with the Pri1 and stably engage the T/P10 and be converted to the polymerase pose”. We have also made several changes in the Discussion paragraph relating to this, as requested, to explain our conclusions in a less definitive fashion.

Minor comments:

1. Many figure citations were incorrect or inaccurate, which should be revised.

Just to name one: Page 4, paragraph 3, “Upon binding template ssDNA, the Pol1-core dissociates from the platform, leading to a drastically different Pol α -primase structure to form the template-bound primer initiation state (Fig. 2b-d, Fig. S3, S8b, Supplementary video 1).” Fig. 2b-d should be Fig. 2d-f. The authors should carefully go through the manuscript to make corrections.

Thanks for catching these errors. We have gone over the full manuscript and checked/updated all figure citations.

2. Page 5 and paragraph 4, “The dumbbell then pivots on the Pri2-CTD and rotates counterclockwise $\sim 40^\circ$ away from the Pol1-core (Fig. 4c)”. The 40° -rotation angle should be labeled in the figure panel. We have now labelled the figure accordingly.

3. Page 6 and paragraph 4, “We have visualized all major intermediate states of a eukaryotic Pol α -primase in this study”. “All” is not appropriate.

Good point. We have revised this to read:

“We have visualized several key intermediate states of a eukaryotic Pol α -primase in this study”.

4. Supporting figures are not cited in order.

We have now checked to ensure that the Supplemental figures are cited in order.

Reviewer #3 (Remarks to the Author):

The manuscript by Li and co-workers presents an impressive structural evaluation of the dynamic activities of yeast DNA Pol alpha by cryoelectron microscopy. The interplay between the subunits has been a subject of interest by many groups since the discovery of a DNA primase activity associated with DNA polymerase alpha in 1982, and this work demonstrates the elaborate orchestration of numerous steps in the process of primer formation followed by DNA polymerization. The manuscript is very well written and interesting, and the figures and video illustrate very effectively the structural findings. The cartoons are especially helpful to visualize the enzymatic process. Though it is difficult for a non-expert in the cryoelectron microscopy field to evaluate comprehensively the structural data collected, the description of the methodology and the data shown in the supplemental figures is thorough and compelling. This reviewer would have appreciated the inclusion of numerous panels of the reconstructed images presented in the main figures alongside the structures derived from the data. At present, only figure 2e shows such a reconstructed micrograph and it is really too small to see and should be presented on the scale of the structures.

We thank the reviewer for the positive comments. The reconstructed images (2D class averages) are shown in **Supplemental Figure 1** for all captured intermediates. There, the main domain movement between consecutive intermediate states are labeled by curved arrows. To further address the reviewer’s concern, we have now incorporated one typical (most informative) averaged image next to each associated 3D map in the revised **Figs. 2-5**. These 2D images not only show the overall architecture but also provide a visual hint on the flexible region(s) in each state. We appreciate the reviewer for taking the time to help us improve our manuscript.

REVIEWERS' COMMENTS

Reviewer #1 (Remarks to the Author):

The authors have addressed all my concerns about the cryo-EM data and models. I believe the manuscript is suitable for publication.

I noticed that, in their rebuttal letter, the authors say: "We have performed additional 3D classification, multibody refinement, and 3D-flex analysis. These have led to appreciable improvement in the partially flexible and weak-density regions". However, no indication of such analyses (multibody refinement or 3D-flex) are mentioned in the revised manuscript. If such analyses have been done and resulted in improved maps, they should be described in the paper.

Reviewer #2 (Remarks to the Author):

My previous comments are mostly on figure presentations. The authors have addressed all my concerns. This is an important work and should be published without further delay.

Reviewer #3 (Remarks to the Author):

I am satisfied with the substantive effort the authors have made to address all of the reviewer queries and recommend publication.

RESPONSES TO REVIEWERS' COMMENTS

Reviewer #1 (Remarks to the Author):

The authors have addressed all my concerns about the cryo-EM data and models. I believe the manuscript is suitable for publication.

I noticed that, in their rebuttal letter, the authors say: "We have performed additional 3D classification, multibody refinement, and 3D-flex analysis. These have led to appreciable improvement in the partially flexible and weak-density regions". However, no indication of such analyses (multibody refinement or 3D-flex) are mentioned in the revised manuscript. If such analyses have been done and resulted in improved maps, they should be described in the paper.

We apologize for our confusing statement. We meant to say in our previous response letter that the extra refinements (multi-body and 3D-flex) only led to marginal and cosmetic improvements, e.g., the slightly improved nominal resolution number, but there was no real and substantive improvement to the final EM 3D maps. For this reason, we decided not to replace the maps in our revised manuscript.

Thank you very much for taking time to help us improve our manuscript.

Reviewer #2 (Remarks to the Author):

My previous comments are mostly on figure presentations. The authors have addressed all my concerns. This is an important work and should be published without further delay.

Thank you very much for taking time to help us improve our manuscript.

Reviewer #3 (Remarks to the Author):

I am satisfied with the substantive effort the authors have made to address all of the reviewer queries and recommend publication.

Thank you very much for taking time to help us improve our manuscript.